# H1: Bootstrapping LLMs to Reason over Longer Horizons via Reinforcement Learning

## Abstract

Large language models excel at short-horizon reasoning tasks, but performance drops as reasoning horizon lengths increase. Existing approaches to combat this rely on inference-time scaffolding or costly step-level supervision, neither of which is scalable. In this work, we introduce a scalable method to bootstrap long-horizon reasoning capabilities using only existing, abundant short-horizon data. Our approach synthetically composes simple problems into complex, multi-step dependency chains of arbitrary length. We then train models on this data using outcome-only rewards under a curriculum that automatically increases in complexity, allowing RL training to be scaled much further without saturating. Empirically, our method generalizes remarkably well: curriculum training on composed 6th-grade level math problems (GSM8K) boosts accuracy on unseen, Olympiad-level benchmarks (GSM-Symbolic, MATH-500, AIME) by up to $2.65\times$. It also transfers significantly to diverse out-of-distribution ReasoningGym domains and long-context benchmarks, indicating broader generalization. Importantly, our long-horizon improvements are significantly higher than baselines even at high *pass@k*, showing that models can learn entirely new reasoning paths under RL. Theoretically, we show that curriculum-based RL with outcome rewards could achieve an exponential improvement in sample complexity over full-horizon training, comparable to the gains from dense supervision, while providing strong training signal without additional human-annotations. *h1* therefore introduces an efficient path towards scaling RL for longer horizons using existing data.

## 1 Introduction

Large language models (LLMs) have improved remarkably in many domains, but they often struggle with long-horizon reasoning (LHR). This involves carrying out a correct, multi-step reasoning process that involves decomposing goals into intermediate steps and executing them successfully in a chain of thought (CoT). Such tasks require reasoning over a sequence of dependent steps where errors can compound across the horizon (Li et al., 2024; Malek et al., 2025; Zhou et al., 2025a; Sinha et al., 2025). For many tasks of interest, such as performing research-level mathematics, debugging complex code, and assisting with scientific discovery, an LLM must be able to correctly solve intermediate problems, carry forward results, and determine what state is important to track and use. Broadly, any hard tasks that are of importance require solving several difficult steps, which motivates the development of training methods directly aimed at improving capabilities on such long sequences of problems.

Reinforcement learning (RL) has shown substantial benefits when it comes to improving the reasoning capabilities of LLMs (OpenAI-o1 et al., 2025; DeepSeek-AI et al., 2025). However, RL depends heavily on the availability of verifiable data and is therefore limited in terms of the complexity of the training data and long-horizon reasoning paths afforded by this data. Moreover, the lack of increasing problem complexity and diversity in RL datasets for LLMs leads to rapidly saturating improvements after a limited number of training steps (Cui et al., 2025; Wu et al., 2025). Obtaining long-horizon training data is expensive and sample inefficient to directly train on (as we discuss in Section 4 and Appendix B). Improving performance on such tasks often requires step-level supervision that is costly, domain specific, and unavailable for most reasoning tasks. Existing approaches (Zhang et al., 2025; Liu et al., 2025b) do not adequately address the problem of improving long-horizon reliability when only short-horizon data is abundant (as is the case in real-world scenarios).

This raises a natural question: **Can we improve long-horizon reasoning capabilities by *scaling reinforcement learning* using only existing short-horizon or single-step training data?**

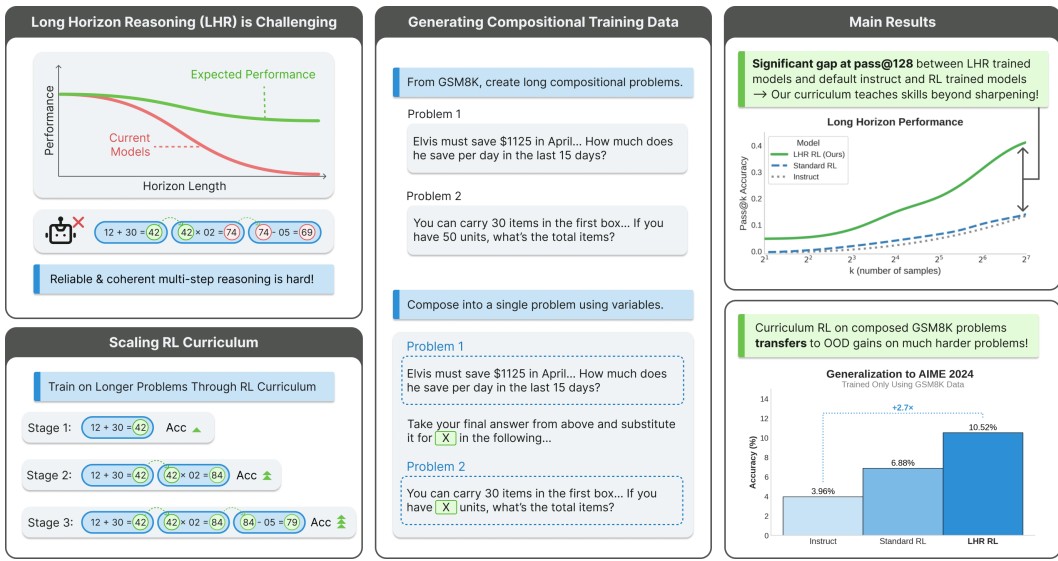

Figure 1: Our approach improves long horizon reasoning by composing existing short-horizon problems into a curriculum for scaling RL training. We observe significant OOD improvements.

In this work, we show that the answer is yes. We introduce a method for chained problem construction, which composes short-horizon problems (e.g. GSM8K problems (Cobbe et al., 2021)) into arbitrarily long chains of dependent reasoning steps. This provides scalable synthetic long-horizon data, with explicit control over the horizon length and complexity without the need for new annotations. We then train language models on this data using reinforcement learning with outcome-only rewards, coupled with a curriculum over horizons. Obtaining useful data that is of just the right complexity for models to learn from has always been a major bottleneck (Wu et al., 2025). We show how existing tasks can be grouped adaptively into increasingly harder problems that provide useful training signal and prevent RL improvements from quickly saturating (Cui et al., 2025). Our approach requires neither step-level labels nor auxiliary models (as in PRMs), and avoids inference-time search, instead directly training the model to internalize long-horizon reasoning structures.

Our results in Sections 4 and 5 show that not only does this synthetic curriculum generalize to other in-domain multi-hop problems, but also transfers to harder benchmarks such as MATH-500 and AIME that implicitly require LHR. Importantly, we show that long-horizon reasoning depends on more than just improving single step accuracy, and provide a breakdown of the capabilities needed for improved performance in Section 3. We evaluate our long-horizon trained models versus other strong baselines up to *pass@128* and show that while improvements obtained from RLVR on standard data is bounded by the base model's capabilities (Yue et al., 2025), our method performs significantly better. This reflects genuinely new skills learnt via curriculum based training on compositional tasks, and we provide an in depth empirical exploration in Section 4 along with robust theoretical results in Appendix B. In Section 6, we further analyze compute–data tradeoffs, showing how scaling compute can substitute for scarce long-horizon data in real-world scenarios.

Our main contributions are:

1. A general method for constructing long-horizon reasoning data by chaining existing short-horizon problems with no additional human-annotations.

2. A reinforcement learning framework with curriculum training and outcome-only rewards that significantly improves horizon generalization and teaches new reasoning paths not elicited otherwise even at very high pass@k.

3. Empirical evidence of transfer to significantly harder benchmarks (MATH-500, AIME, GSM Symbolic, LongBench-v2, Hash-hop) while training on compositional GSM8K data.

4. Generalization to out-of-distribution multi-hop reasoning domains from ReasoningGym.

5. Theoretical analysis of sample complexity of curriculum learning, showing that it could achieve an exponential improvement over full-horizon training, similar to dense rewards.

## 2 RELATED WORK

**LLM Reasoning and RL.**  Initial reasoning literature (Zelikman et al., 2022) bootstrapped performance using model generated reasoning traces. More recently, (OpenAI-o1 et al., 2025; DeepSeek-AI et al., 2025) demonstrated substantial improvements in reasoning capabilities via RL training. These advances have enabled effective scaling of inference-time compute (Snell et al., 2024; Brown et al., 2024; Muennighoff et al., 2025). However, as reasoning chains grow longer, models exhibit several limitations, often struggling with simple multi-step problems (Malek et al., 2025; Shojaee et al., 2025; Song et al., 2025). Moreover, RL-based approaches face their own challenges: diversity degradation during training (Song et al., 2025), questions about whether models truly acquire new capabilities versus better sampling existing ones (Yue et al., 2025), and maintaining stability over long horizons (Xiang et al., 2025). Recent efforts toward addressing these challenges include Setlur et al. (2025), which improves in-context exploration via an RL curricula with steps such as verification and refinement and work on adaptive difficulty scheduling for efficient training (Shi et al., 2025; Parashar et al., 2025; Liu et al., 2025b). Xi et al. (2024) collect a dataset that requires step-level demonstrations and RL post-train on this fixed dataset by following a curriculum. In contrast, our work systematically composes existing short-horizon problems into chains of increasing length, producing new data to scale an RL curriculum to train models to internalize long-horizon reasoning capabilities that they otherwise lack. This enables reliable multi-step problem solving and improvements on significantly harder (unseen) settings, providing a foundation for training long-horizon agents (Zhou et al., 2025b; Kwa et al., 2025) that can track complex state and execute dependent reasoning steps over extended sequences.

Additionally, a detailed discussion of our work and its novelty in the context of length generalization and long-context models is provided in Appendix Section A.

## 3 METHOD

Long-horizon reasoning refers to the capability of carrying out a coherent, multi-step reasoning process and executing steps reliably in a CoT to solve long horizon tasks.

**What counts as a long-horizon task?**  We use two notions. *Explicit-horizon* tasks have a known number of dependent sub-problems $h$ because we construct them by chaining atomic problems (used for training and in-domain evaluation). *Implicit-horizon* tasks require multiple dependent reasoning steps but do not come with an explicit decomposition (e.g., MATH-500, AIME); they have a latent horizon $h^\star$ that is not annotated. Our training targets explicit horizons for clean analysis, and shows a strong transfer to implicit-horizon benchmarks.

Our goal is to *bootstrap* long-horizon reasoning (LHR) using only existing short-horizon data. We (i) compose atomic problems into longer chains of problems with dependent steps to synthesize LHR data, (ii) scale RL training with outcome-only GRPO following a curriclum learning approach, and (iii) evaluate both in-domain (explicit chains) and on harder out-of-domain tasks that implicitly require many reasoning steps. Here, we describe what we mean by a long-horizon tasks, formalize our data construction process, and provide details about our RL training objective.

**Atomic tasks and serial composition.**  We begin with *atomic tasks* $f_j$: short, self-contained problems (e.g., single GSM8K questions) with verifiable answers that the base model solves with non-trivial accuracy. Each task takes an input $x_j$ and produces an answer $y_j$.

To form long-horizon examples, we chain $h$ atomic tasks so later sub-problems depend on earlier results. A lightweight *adapter* $\phi_j$ maps $y_j$ to the next input:

$$y_j = f_j(x_j), \qquad x_{j+1} = \phi_j(y_j), \qquad j = 1, \ldots, h-1,$$

yielding the final answer

$$y_h = f_h(\phi_h(\cdots \phi_1(f_1(x_1)))).$$

Adapters may be identity or simple deterministic transforms (e.g., scaling, unit conversion). Each chain of length $h$ is rendered as a single prompt listing the $h$ sub-problems in order. The model is instructed to solve them sequentially but is supervised only on the final answer $y_h$ (outcome-only RL). We apply basic well-posedness checks (type/range consistency, unit compatibility, de-duplication).

---

**Example explicit-horizon chain**

**1.** Weng earns \$12 an hour for babysitting. Yesterday, she babysat for 50 minutes. How much did she earn? (#1)

**2.** Betty is saving money for a new wallet which costs $\${10 \times \#1}$. Betty has only half of the money she needs. Her parents give her \$15, and her grandparents give her twice as much as her parents. How much more money does Betty need to buy the wallet? (#2)

**3.** James writes a $\{\#2\}$-page letter to 2 different friends twice a week. How many pages does he write a year? (#3)

---

This construction exposes models to dependency chains that require carrying, transforming, and reusing intermediate values, while keeping supervision outcome-only. We vary chain length $h$ to implement the stagewise curriculum described later in this section. In Appendix A.1, we analyze our composition method through computational graphs to explain its effectiveness during training.

**Why horizons are hard: beyond multiplicative errors.** In *explicit-horizon* tasks, let $h$ be the number of dependent sub-problems whose intermediate values are reused downstream. An independent-errors view gives $P(\text{final correct}) = p^h$, suggesting that raising atomic step accuracy $p$ suffices. This is *overly optimistic* because it ignores *context management*: as transcripts grow, models can lose or corrupt intermediate values even when each step is easy. We model long-horizon accuracy via *atomic reliability* $p$ and *context management* $\sigma_j$ (the chance the required information is correctly retrieved at step $j$). Writing $s_j$ for the probability that the reasoning state remains correct after step $j$, we have

$$s_j = p\,\sigma_j\,s_{j-1}, \qquad s_0 = 1,$$

so if $\sigma_j$ decays with horizon length, accuracy can collapse even when $p \approx 1$.

This explains the weakness of naive outcome-only training at horizon $h$: when $\sigma_j \ll 1$, few rollouts earn reward, gradients have low signal-to-noise ratio, and samples scale exponentially in $h$. Curriculum training mitigates this by starting with short chains where $s_j$ is large, yielding high-SNR updates; early stages raise $p$, while later stages reinforce write/read behaviours that stabilise $\sigma_j$. Empirically (Section 4), performance depends on capabilities beyond $p$, and our approach improves both $p$ and $\sigma_j$, delivering large gains on *explicit-horizon* tasks and generalising to harder *implicit-horizon* tasks (Section 5); Appendix Section B develops the theoretical implications.

---

**Algorithm 1** h1: Stagewise curriculum RL over explicit-horizons

---

**Require:** Pretrained model $M_0$; atomic task bank $\mathcal{A}$; adapters $\{\phi_j\}$; max horizon $H_{\max}$; per-stage counts $M_h, S_h$

1: **for** $h = 1$ to $H_{\max}$ **do**               ▷ stagewise curriculum over explicit horizons
2:     $\mathcal{D}_h \leftarrow \emptyset$
3:     **for** $m = 1$ to $M_h$ **do**               ▷ construct horizon-$h$ chains
4:        sample $(f_{1:h}, x_1)$ from $\mathcal{A}$;    $y_1 \leftarrow f_1(x_1)$
5:        **for** $j = 1$ to $h - 1$ **do**
6:           $x_{j+1} \leftarrow \phi_j(y_j)$;   $y_{j+1} \leftarrow f_{j+1}(x_{j+1})$
7:        $p \leftarrow \text{FORMATPROMPT}((f_j, x_j)_{j=1}^{h})$         ▷ format prompt from the task sequence
8:        $\mathcal{D}_h \leftarrow \mathcal{D}_h \cup \{(p_{1:h}, y_h)\}$
9:     $M_h \leftarrow \text{TRAINWITHDRGRPO}(M_{h-1}, \mathcal{D}_h, S_h)$

---

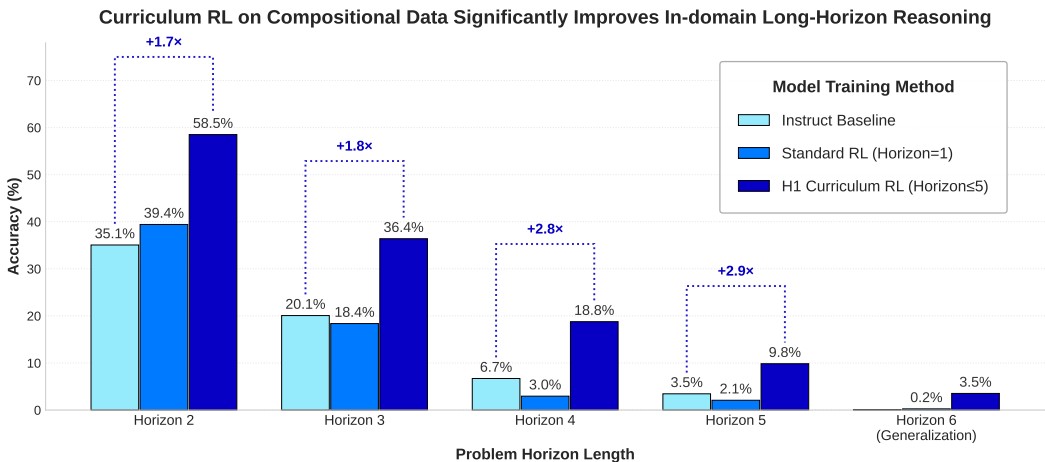

Figure 2: Curriculum RL training on compositional data offers significant in-domain long horizon reasoning gains (**up to 2.9×**). This prevents RL training from saturating and uses no new data.

**Scaling RL with a curriculum over horizons.**    Let $\mathcal{D}_h$ be the dataset of synthesized chains of explicit horizon $h$. Our curriculum is stagewise:

for $h = 1, 2, \ldots, H_{\max}$ :     run DrGRPO (Liu et al., 2025c) on $\mathcal{D}_h$ for $S_h$ optimization steps.

We initialize from $\pi_{\theta_0}$ and carry the parameters forward between stages. Algorithm 1 describes the entire training process. By focusing optimization on a single horizon per stage, the model first acquires reliable short-horizon primitives (increasing $p_1$), then learns to reuse and repair them under longer dependency (increasing $p_j$ and $r_j$ for $j > 1$). We contrast the curriculum with three baseline horizon-sampling policies:

1. **Only-L1:** $q(\ell) = \mathbb{I}[\ell = 1]$. If direct problem-solving were sufficient, this would match curriculum; empirically it does not.

2. **Uniform-Mix:** $q(\ell) \propto \mathbb{I}[1 \le \ell \le H_{\max}]$, i.e., randomly pick from the LHR dataset.

3. **Only-Long:** $q(\ell) = \mathbb{I}[\ell = H_{\max}]$, i.e., train solely on the hardest chains. This suffers from extreme sparsity and unstable gradients.

Generally, RL with verifiable rewards (RLVR) requires the creation of a clean labeled dataset. What models can learn from is potentially limited by the complexity expressed in these problems. We see this bound due to a fixed RL dataset both empirically (Section 3) and theoretically (Appendix B), which leads to performance quickly saturating during training. Our goal with a synthetic curriculum is to optimally utilize limited existing data for scaling RL. At each stage, tasks can be composed to be right at the edge of what a model can solve, making RLVR more scalable (see Tables 1 and 2).

**Training and evaluations.**    We use the Qwen-2.5-3B Instruct model (Qwen et al., 2025) for our core experiments. Improving an Instruct model with RL is generally considered more difficult (Wang et al., 2025) and gains signify performance improvements beyond just instruction tuning (which cannot be directly inferred for improvements on base models (Shao et al., 2025)). Therefore, we aim to show all improvements on Instruct models for the purpose of robustness. Our *explicit-horizon* training and evaluations are done on composed GSM8K questions (Cobbe et al., 2021), and our *implicit-horizon* evaluations are on AIME 2024, AIME 2025, MMLU Pro Math (Wang et al., 2024), GSM Symbolic (Mirzadeh et al., 2025), and MATH-500 (Hendrycks et al., 2021).

## 4    IN-DOMAIN RESULTS AND THE IMPORTANCE OF CURRICULUM

We evaluate our curriculum-based RL training method using explicit-horizon GSM8K problems and demonstrate that **(1)** curriculum learning is essential for long-horizon reasoning, **(2)** LHR performance depends on capabilities beyond single step accuracy, and **(3)** our method teaches genuinely new capabilities that are otherwise absent in the model. We use **Qwen-2.5-3B Instruct**

| Model / setting | Accuracy on GSM8K Problems of Horizon L-$n$ | | | | | | | |
|---|---|---|---|---|---|---|---|---|
| | L-1 | L-2 | L-3 | L-4 | L-5 | L-6 | L-7 | L-8 |
| **Instruct model** | 82.79 | 35.06 | 20.07 | 6.70 | 3.57 | 0.00 | 0.79 | 0.00 |
| *Equal compute training baselines* | | | | | | | | |
| **Only-L1** | 86.80 | 37.14 | 21.43 | 6.70 | 3.87 | 0.25 | 0.00 | 0.00 |
| **Uniform-Mix** | 82.80 | 12.66 | 2.04 | 0.54 | 0.00 | 0.00 | 0.00 | 0.00 |
| **Only-Long** | 82.71 | 43.36 | 20.41 | 3.22 | 1.49 | 0.25 | 0.25 | 0.00 |
| *Increased Inference Compute Baseline* | | | | | | | | |
| **Tree of Thought** | 83.30 | 39.40 | 13.30 | 2.00 | 0.00 | 0.00 | 0.00 | 0.00 |
| *Curriculum training (trained up to Len-$n$)* | | | | | | | | |
| **RLVR** | 83.20 | 39.42 | 18.37 | 2.95 | 2.08 | 0.25 | 0.79 | 0.00 |
| **Len-2** | 85.92 | 56.22 | 28.57 | 12.06 | 6.25 | 1.26 | 0.79 | 0.49 |
| **Len-3** | 84.91 | 56.22 | 37.76 | 15.55 | 8.63 | 3.27 | 3.17 | 0.25 |
| **Len-4** | 85.48 | 57.05 | **40.14** | 18.23 | 9.23 | 3.53 | 3.17 | 1.72 |
| **Len-5 (H1)** | **85.97** | **58.51** | 36.39 | **18.77** | **9.82** | **3.53** | **3.17** | **2.22** |
| | (+3.8%) | (+66.9%) | (+81.3%) | (+180.1%) | (+175.1%) | (++) | (+301.3%) | (++) |

Table 1: GSM8K accuracy by horizon length. Curriculum based RL training **significantly improves** in-domain performance compared to the Instruct model and all other equal compute baselines. We also provide a Tree-of-Thought (Yao et al., 2023) baseline using Qwen-2.5-3B-Instruct.

for our experiments, with GRPO over a curriculum of chained GSM8K problems with horizons $h \in \{1, 2, 3, 4, 5\}$. Each stage trains for 200 steps with 200 samples per horizon. We compute the following baselines: **Only-L1** (standard RL on $h=1$), **Only-Long** ($h=5$), and **Uniform-Mix** (uniform over $h \in [1, 5]$). Compute matched baselines are trained using up to the same number of training tokens seen under our method, and the best checkpoints are chosen based on val-set accuracy. The data comes from the same training distribution (where Only-L1 refers to simply training on horizon 1 problems, Only-Long refers to training only on composed horizon 5 problems, and Uniform-Mix refers to training on the same problems as our method but shuffled uniformly without a curriculum). In Appendix D, we provide results on *Qwen-2.5-7B Instruct* using composed MATH and *Llama-3.2-3B Instruct* using composed GSM8K data, both showing improvements.

**In-domain results.** In Table 1, our in-domain results on composed LHR GSM8K problems from the test set show that the curriculum-based approach yields substantial monotonic improvements in accuracy as the training horizon increases. At $h=2$ the instruct model achieves 35.06%, which increases to only 39.42% with RL on standard GSM8K problems but jumps to 56.22% when training up to a horizon of 2 and 58.51% when trained up to a horizon of 5. Similarly, at $h=3$ the instruct model achieves 20.07%, which lifts to 37.76% with a curriculum up to $h=3$. For longer horizons (harder problems), the effect of curriculum is even more visible, increasing accuracy by about $3\times$ at $h=4$ (6.70% $\rightarrow$ 18.77%) and $h=5$ (3.57% $\rightarrow$ 9.82%). We present these improvements in Figure 2.

In Table 1, the **Only-L1** baseline improves $h=1$ but shows no improvements on longer horizons. Similarly, **Uniform-Mix** even at an equal training compute baseline shows no improvements. **Only-Long** also leads to no long-horizon improvements due to the lack of useful training signal at longer lengths discussed in Section 3. Furthermore, Cui et al. (2025) show that the entropy of a policy undergoing RL training collapses quickly, which causes improvements from RL to saturate quickly. While this is true for our baselines, our curriculum training repeatedly introduces new levels of difficulty (exploration), which allows *scaling RL for up to 5x more steps* to keep improving capabilities. We leave a deeper investigation into the scaling properties of our method to future work.

> **Curriculum RL bootstraps long-horizon reasoning**
>
> Training up to horizon $h$ extends usable learning signal on $h + 1$ and shifts probability mass into the long-sequence tail monotonically. For e.g. training to $h=3$ lifts $h=4$ from 6.70%

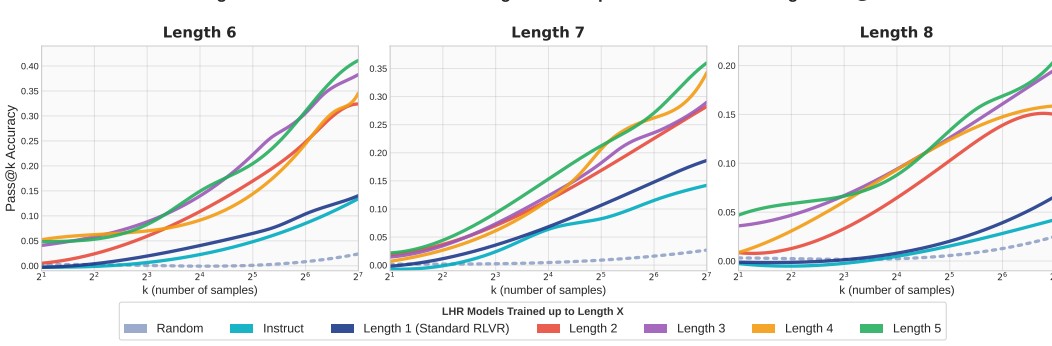

Figure 3: Our curriculum based RL training using composed synthetic data outperforms RLVR on standard data from the same set even at *pass@128*, **teaching new capabilities that did not previously exist in the base model**. LHR requires going beyond improving single-step performance.

> to $15.55\%$; training to $h=4$ lifts $h=5$ from $3.57\%$ to $9.23\%$. This provides enough training signal for the next stage, allowing curriculum learning to be extremely effective. We examine this theoretically in Appendix B.

**Why single-step accuracy is not enough.** In Section 3, we claim that LHR depends on more than just single step accuracy. Prior to RL training, single-step accuracy of the model is $82.79\%$. If errors were independent, we would expect $68.54\%$ at $h=2$ and $56.75\%$ at $h=3$ by multiplicative compounding, yet we observe $35.06\%$ and $20.07\%$ (Table 1). Even after RL training (**Only-L1**) for 200 steps, (despite a slight increase at $h=1$) performance drops to $39.42\%$ at $h=2$ and $18.37\%$ at $h=3$ rather than $69.28\%$ and $57.67\%$ expected under the independent error assumption.

**Learning new capabilities with RL.** We now discuss the second part of our claim in Section 3. LHR depends on additional capabilities such as state tracking and repair that can be improved using RL training over a curriculum. (Yue et al., 2025) show an important result that RLVR on LLMs only improves the sample efficiency of reasoning capabilities already present in the base model, and no new capabilities are learnt. They show that at a high pass@k (such as 128), capabilities of these RL trained models originate from and are bounded by the base model (with the pass@k performance quickly converging). Therefore, only when an RL model is not bounded by the base model at high pass@k can one empirically show new capabilities are learnt.

Our *explicit-horizon* training and testing setting allows us to isolate out these capability improvements that go beyond the base model with only RL. Importantly, proving one of the central claims in our paper, we evaluate our final model on unseen longer horizons ($h = 6, 7,$ and $8$) up to a very high sampling budget *(pass@128)*. Our results in Figure 3 show that while RL on standard GSM8K is bounded by instruct model capabilities (and converges very quickly), our long horizon trained models perform significantly better even at high $k = 128$. This shows our method unlocks new, correct reasoning paths that were previously inaccessible to the model, providing genuinely new LHR capabilities. This is a significant finding compared to common RLVR training paradigms studied in (Yue et al., 2025), showing that our RL method can indeed teach new reasoning skills when training.

> **LHR Training can teach new capabilities**
>
> We demonstrate for the first time that **Curriculum RL can teach new capabilities that go significantly beyond the base model even at *pass@128***. Our curriculum based training on compositional synthetic data is therefore crucial.

In this section, we show significant improvements on explicit-horizon in-domain tasks and that our model learns new reasoning capabilities with our curriculum based training. Our explicit-horizon GSM8K setting, while very useful in allowing us to isolate these capabilities and understand the

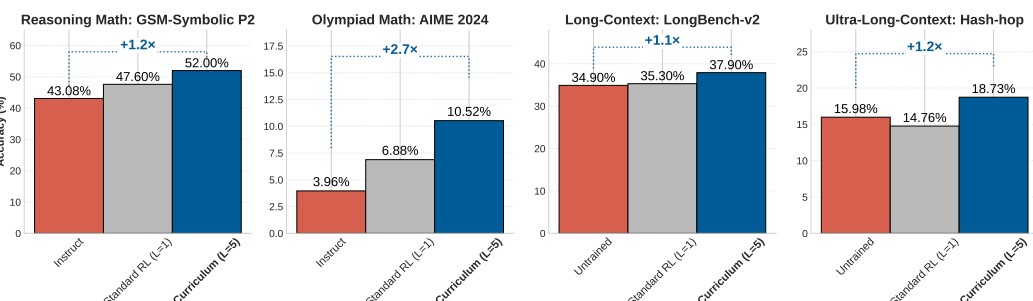

Figure 4: Long-horizon training on GSM8K generalizes to significantly harder tasks. **Performance on AIME 2024 improves by $2.65\times$ and ultra-long-context capabilities improve by $1.2\times$.**

differences in all training methods, is still relatively artificial. In Section 5, we therefore test our GSM8K trained LHR models on significantly harder (unseen) problems.

## 5 GENERALIZATION TO HARDER BENCHMARKS

Having established that our curriculum-based training imparts new, in-domain capabilities, we now investigate whether these learned skills generalize to challenging, out-of-domain benchmarks that implicitly require long-horizon reasoning. Our results (Figure 4) demonstrate that the skills acquired from solving synthetically chained problems transfer remarkably well to harder problems.

**Transfer to Olympiad level math.** In Table 2, we evaluate our GSM8K long horizon trained models on MATH-500, GSM-Symbolic P1, GSM-Symbolic P2, MMLU Pro Math, and AIME. These tasks can be categorized as *implicit-horizon* and benefit significantly from LHR training on much easier *explicit-horizon* tasks. For instance, performance on GSM-Symbolic P1 goes from $67.06 \rightarrow 73.28$, P2: $43.08 \rightarrow 52.00$), and strikingly AIME 2024 from $3.96 \rightarrow 10.52$, a $2.65\times$ increase. These improvements show a transfer of the capabilities targeted in Section 4.

| Model/setting | **Generalization to Significantly Harder Math Problems** | | | | | |
|---|---|---|---|---|---|---|
| | **MATH-500** | **Symbolic P1** | **Symbolic P2** | **MMLU-Pro** | **AIME 2025** | **AIME 2024** |
| **Instruct model** | 64.20 | 67.06 | 43.08 | 58.47 | 1.77 | 3.96 |
| *Standard RLVR on GSM8K* | | | | | | |
| **GSM8K RLVR** | 66.20 | 71.40 | 47.60 | 60.62 | 2.71 | 6.88 |
| *Equal compute training baselines* | | | | | | |
| **Only-L1** | 48.40 | 71.32 | 42.24 | 59.84 | 3.12 | 5.31 |
| **Uniform-Mix** | 64.40 | 64.48 | 39.16 | 60.22 | 2.50 | 5.28 |
| **Only-Long** | 65.60 | 72.18 | 47.52 | 60.71 | 1.72 | 6.46 |
| *Curriculum RL on Composed GSM8K Problems* | | | | | | |
| **Len-2 GSM8K** | 67.00 | 72.86 | 50.80 | 59.73 | 1.25 | 4.69 |
| **Len-3 GSM8K** | 66.80 | 70.70 | 49.48 | 61.21 | 1.67 | 3.85 |
| **Len-4 GSM8K** | 68.40 | 72.22 | 51.92 | 60.91 | 2.60 | 7.60 |
| **Len-5 GSM8K** | **69.20** | **73.28** | **52.00** | **61.21** | **3.02** | **10.52** |
| | (+7.8%) | (+9.3%) | (+20.7%) | (+4.7%) | (+70.6%) | (+165.7%) |

Table 2: Performance on harder math benchmarks improves significantly with GSM8K RL curriculum training stages. Bootstrapping simple existing data can be used for scaling RL. AIME avg@32.

LHR training allows us to bootstrap capabilities from significantly easier tasks to gains on much harder ones without using any extra labels or supervision. We see a scaling trend, where continued

RL training on longer *explicit-horizons* leads to improvements on *harder implicit-horizon* tasks. Bootstrapping composed LHR data can allow more RL compute to be spend on the same dataset.

> **Generalization to Olympiad Level Problems**
>
> Training on composed 6th grade problems with our RL curriculum generalizes to significantly harder benchmarks. Notably, we achieve a **2.65× improvement on AIME 2024.**

**Transfer to long-context benchmarks.** We now evaluate our GSM8K LHR models on OOD long-context benchmarks to see if the state tracking capabilities ($\sigma_j$) from Section 3 improve. We test two main long-context benchmarks: LongBench-v2 (Bai et al., 2025) and Hash-hop (Magic, 2024). LongBench-v2 measures understanding and reasoning over QA documents, long-dialogue, repositories, etc. (with 8k–2M words). Hash-hop tests ultra-long-context storage, retrieval, and multi-hop variable tracing by making models follow shuffled chains of random hash $\rightarrow$ hash pairs. Table 6 summarizes our results, with a $35.00\% \rightarrow 37.90\%$ improvement on LongBench-v2 and a $15.98\% \rightarrow 18.73\%$ improvement on Hash-hop, both completely unrelated to GSM8K.

**Transfer to non-mathematical reasoning benchmarks.** We also test our long-horizon trained models on ReasoningGym (Stojanovski et al., 2025) domains to evaluate whether the *horizon-dependent reliability* improvements generalize to non-mathematical but verifiable reasoning tasks. ReasoningGym consists of a diverse set of reasoning environments that allow us to evaluate cross-domain transfer and skill generalization. Specifically, we test across logic (propositional logic), graphs (largest island), algorithmic problems (sentence reordering and matrix manipulation), arithmetic (decimal arithmetic), and geometry. These problems require working memory, graph traversal, multi-step rule following, and correct final answers. On ReasoningGym, long-horizon training on composed GSM8K significantly outperforms both the Instruct model and RLVR trained on normal GSM8K. $h1$ generalizes from $22.90\% \rightarrow 47.10\%$ on propositional logic, $15.00\% \rightarrow 22.50\%$ on graph problems (largest island), $9.60\% \rightarrow 18.80\%$ on algorithmic sentence reordering, and $2.70\% \rightarrow 4.20\%$ on algorithmic matrix manipulation. Performance on geometry drops from $3.70\% \rightarrow 2.60\%$ and on games (game of life) from $76.20\% \rightarrow 74.90\%$. Overall, skills learnt from long-horizon training generalize well to out-of-distribution reasoning problems. See Table 3.

| Model / setting | Generalization to ReasoningGym domains | | | | Long-Context Benchmarks | |
|---|---|---|---|---|---|---|
| | Propositional logic | Graphs (largest island) | Algorithmic (sentence reorder) | Algorithmic (matrix) | LongBench-v2 | Hash-hop |
| Instruct | 22.90 | 15.00 | 9.60 | 2.70 | 35.00 | 15.98 |
| Standard RLVR | 12.40 | 17.00 | 9.80 | 3.90 | 35.30 | 14.76 |
| Long-horizon RL | **47.10** | **22.50** | **18.80** | **4.20** | **37.90** | **18.73** |

Table 3: Long-horizon training on composed GSM8K problems generalizes remarkably well to OOD *ReasoningGym* domains and *Long-Context Benchmarks*, outperforming length-1 (standard) RLVR and the Instruct model. We use default ReasoningGym configurations for our evaluations.

**Analysis.** This transfer patterns aligns with our *pass@k* capability improvement results from Section 4 and our theoretical framing. Tasks requiring sequential dependent reasoning, such as AIME or GSM-Symbolic problems, benefit from improved long-horizon reasoning capabilities that were learned on much simpler composed tasks. Crucially, improvements in aspects such as state-tracking ($\sigma_j$) are also observable from our long-context evals. Our results indicate that a curriculum of simple explicit-horizon tasks can bootstrap advanced reasoning, providing a scalable path where composing problems at the edge of what can be solved would push capabilities further without new annotations.

## 6    DESIGNING A COST EFFICIENT CURRICULUM

In most real-world scenarios, there is an abundance of short-horizon data, and long-horizon data is expensive to obtain (Kwa et al., 2025). In this section, we ask whether long-horizon performance can be obtained from training data distributions that are "cheaper" than a uniform one. Namely,

Figure 5: **Left**: Sample count distributions for four settings. **Middle**: Comparing accuracy at each stage across sample count settings. Under mild skew towards shorter samples like Setting 1 and 2, the model can perform as well as the uniform sample baseline. **Right**: Comparing the training compute across settings. The settings skewed towards shorter samples have more training cost in terms of training tokens seen. Overall, low-cost data distributions can achieve near-optimal performance.

whether we can train on more short data and less long data and still achieve the same performance. We also evaluate how much this changes the training compute required.

Our experiments follow the same curriculum RL method described in Sections 4 and 3. During training, we train up to saturation for each stage, spending as much training compute as needed until there are no further improvements in accuracy. We track the total number of tokens seen by the model. We create three different curricula with the same total number of samples, and different proportions of short- and long-horizon data (Figure 5 left).

The results in Figure 5 (middle and right) show that high long-horizon performance can be achieved even in data-constrained scenarios with training data distributions skewed towards shorter examples, but the trade-off is that we need to spend more training compute overall. However, as seen in the case of Setting 3 (Figure 5, left), a reasonable amount of long horizon data is still needed, otherwise optimal performance may be unreachable. Therefore, to further study this trade-off, we simplify our experimental setup to the SFT setting on a simpler task (multiplication), and scale up the search space for comprehensive evaluations. In Appendix C we provide results that show, for a target accuracy, a similar trade-off exists between (1) training cost and (2) training compute budget.

## 7 DISCUSSION

In this paper, we introduced a novel framework for improving long-horizon reasoning in large language models. Our method leverages existing short-horizon data by constructing new, multi-step problems through a chaining process. This approach allows us to scale reinforcement learning training, yielding substantial performance gains on multi-step reasoning tasks. An important result of our work is that the skills learned through this curriculum transfer to new challenging reasoning and long-context tasks. Furthermore, our results show that the model learns genuinely new reasoning capabilities, rather than just refining existing ones. We demonstrate that comparable performance can be achieved even when there is abundant short-horizon data but limited long-horizon data, thus providing a scalable and data-efficient path for improving frontier models.

While the goal of our paper was to introduce an early method for improving long-horizon reasoning, we see two promising directions for extensions. One is incorporating new sources of atomic skills beyond GSM8K. The other is creating new chaining methods that expands the serial dependency structure in our current method. We believe these two paths would offer useful extensions to the method we introduce in this paper and further improve long-horizon reasoning.

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

## A    ADDITIONAL RELATED WORK.

**Length Generalization.**    Length generalization is concerned with extrapolating to longer sequence lengths than those seen during training (Dubois et al., 2019; Hupkes et al., 2020; Newman et al., 2020; Anil et al., 2022). Length generalization settings mostly focus on small scale tasks (Sabbaghi et al., 2024; Zhou et al., 2024) but do not address RL training of reasoning models. A close example (Lee et al., 2025) uses curriculum construction and SFT to train small transformers on progressively harder algorithmic tasks. In this work, we not only show progressive length generalization gains through curriculum based RL, but also cross-task generalization on much harder tasks.

**Long Context Models.**    Another related thread is extending LLM context length to handle very large inputs. Recent models feature context windows of tens or hundreds of thousands of tokens (Liu et al., 2025a) and benchmarks like LongBench-v2 (Bai et al., 2025) evaluate performance on extremely long inputs such as documents and code. Frontier models with state-of-the-art context windows still suffer performance degradation when required to infer against distant pieces of information or a series of dependent tasks (Li et al., 2024; Malek et al., 2025; Zhou et al., 2025a). These works show that simply having larger context windows does not guarantee that models can perform deep, dependent reasoning over several steps. Our work aims to address this gap by focusing on training for improved long-horizon output generation rather than just long input handling.

### A.1    COMPOSITION AND COMPUTATIONAL GRAPHS.

**Synthetic LHR data construction.**    Let $\mathcal{D}_1 = \{(x, y = f_a(x))\}$ be solved atomic problems (e.g., GSM8K). We build horizon-$h$ examples $(p_{1:h}, y_h)$ in two interchangeable ways:

1. **Transformation chaining.** Given $(x_j, y_j)$, define $x_{j+1} = \phi_j(y_j)$ via a typed, deterministic transformation (e.g., unit conversion, affine reparameterization, substitution into a template). This yields $x_1 \mapsto y_1 \mapsto \cdots \mapsto y_h$ with $y_h$ computed exactly by composition.

2. **Recompute chaining.** Draw an independent atomic instance $\tilde{x}_{j+1}$ and re-compute its key parameters as functions of $y_j$ (e.g., replace a placeholder with $y_j$), producing $x_{j+1} = \psi_j(\tilde{x}_{j+1}, y_j)$ while preserving the solver $f_{a_{j+1}}$.

We render the chain as a single prompt

$$p_{1:h} = R_{a_1}(x_1) \,\|\, R_{a_2}(x_2) \,\|\, \cdots \,\|\, R_{a_h}(x_h),$$

instructing the model to solve the $h$ dependent sub-problems sequentially and return the final answer $y_h$.[1]

We can analyze our method from the perspective of computational graphs (Zhou et al., 2025a). Each verifiable problem, such as in GSM8k dataset, forms a single-sink directional acyclic graph where each node represents an operator consuming the value from previous nodes and producing the value for the next node(s) or as an output of the graph as shown in Figure 6 for an example GSM8k problem. The height of the graph then can represent the number of steps that must be crafted and accurately carried out while the width of the graph represents the state that must be maintained at each step and accurately manipulated. This framing enables us to visualize various possible compositions of a given set of problems. The sequential composition presented in this paper forms a simple composition technique that enables models to learn through a curriculum crafting and evaluating of larger and larger number of the steps that a problem may require.

We generated the computational graphs for all of the problems in GSM8k and AIME24 datasets to examine the patterns of computation and compositions for these problems. While the graphs for these two datasets are not equivalent as they use different operators, they give some insights into why our method is able to show generalization across different datasets. For GSM8k problems, we found the computational graphs have average width and height of 4.1 and 4.0 respectively, while AIME graphs have average width and height of 6.6 and 7.1 respectively. We then compare the statistics for the graphs of AIME problems solved before and after our procedure as shown in table 4. We observe that our procedure enables models to learn creating and evaluating longer computational graphs through sequential composition and curriculum learning.

---

[1]We apply standard well-posedness filters: type checks, numeric range clipping, and de-duplication.

Albert is wondering how much pizza he can eat in one day. He buys 2 large pizzas and 2 small pizzas. A large pizza has 16 slices and a small pizza has 8 slices. If he eats it all, how many pieces does he eat that day?

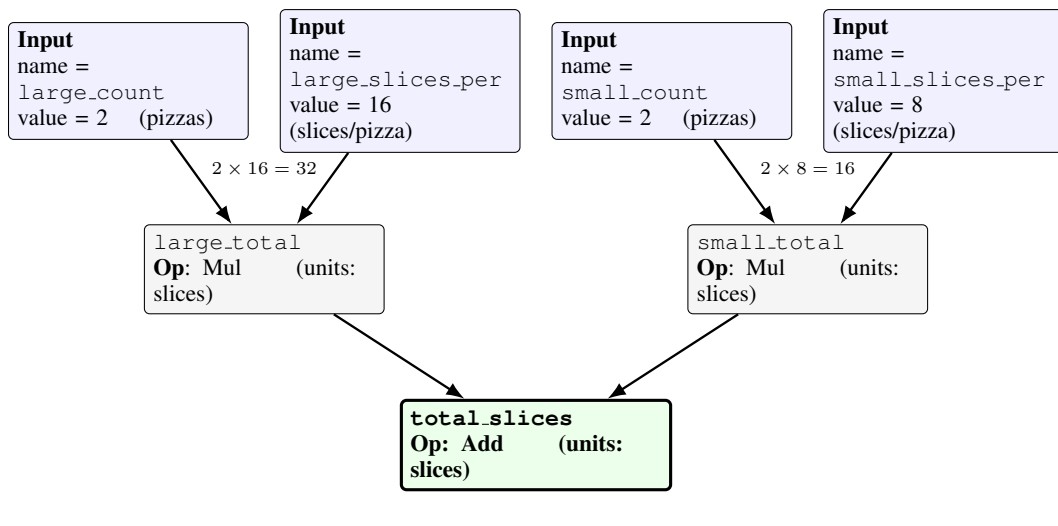

Figure 6: Question and computational graph for a GSM8K problem (final answer: **48**).

Table 4: AIME24 solved problems comparison. N+E denotes Nodes + Edges.

|  | N+E | Width | Height |
|---|---|---|---|
| Baseline: Instruct-model solved problems | 47.25 | 8 | 6.75 |
| Ours: Additional newly solved problems | 54.3 | 4.7 | 10.3 |

## B  THEORETICAL ANALYSIS

Intuitively, under our long-horizon skill model in Section 3, attempting to train directly on long-horizon data with outcome rewards results in vanishing gradient signal, as very few rollouts achieve any reward. Curriculum training overcomes this by initially training at short horizons, where this signal is stronger. Raising the success rate in achieving a reward at horizon $j$ also raises the success rate for horizons $> j$, and so when we come to train at $j + 1$, the signal is no longer vanishing. In the analysis below, we prove that this is the case in our simplified long-horizon skills model, and demonstrate an exponential decrease (with respect to the horizon length $H$) in the sample complexity for curriculum training vs direct outcome reward-only horizon $H$ training, along with an equivalence between curriculum training and training at horizon $H$ with dense, per-step rewards.

### B.1  SETUP AND NOTATION

We study our simplified model of skill acquisition described in Section 3 to analyze the benefits of curriculum learning for LHR. We consider a finite-horizon episodic problem with maximum horizon

$H$, where the probability of being correct up to depth $i$ is

$$s_i = \prod_{j=1}^{i} q_j, \qquad q_j = p(\theta_0)\,\sigma_j(\theta_j) \in (0,1), \quad s_0 = 1,$$

with depth-$j$ parameters $\theta_j$. $p \in (0,1]$ represents homogeneous *atomic task reliability*, while $\sigma_j \in [0,1]$ represents heterogeneous *context length dependent reliability* (we assume $\sigma_1 = 1$). At initialization, we assume $q_j \in [\delta, 1-\delta]$ for some constant $\delta > 0$.

Note that our model does not allow for the possibility of self-correction or error cancellation, and so can be viewed as a simplified analysis of the worst-case average sample complexity for all training regimes. Incorporating these factors into our model would reduce sample complexity in all scenarios, but is likely to leave the exponential separation intact.

We use unbiased advantage-based policy gradient with a leave-one-out (LOO) baseline. For depth $i$, we draw $N$ i.i.d. trajectories $\{y_g\}_{g=1}^N$ with terminal reward $R_i(y_g) \in \{0,1\}$ and advantage

$$A_g = R_i(y_g) - \frac{1}{N-1}\sum_{h \neq g} R_i(y_h).$$

For a block $k \leq i$ the (blockwise) score and estimator are

$$Z_{k,g} = I_{k-1}(y_g)\nabla_{\theta_k}\log\pi_k(y_g) = I_{k-1}(y_g)\frac{B_k(y_g)-q_k}{q_k(1-q_k)}\nabla_{\theta^{(k)}}q_k,$$

$$\bar{g}_k = \frac{1}{N}\sum_{g=1}^N A_g\,Z_{k,g},$$

where $I_{k-1}$ is the reach indicator for step $k$ and $B_k \sim \mathrm{Bernoulli}(q_k)$ is the step-$k$ success. We abbreviate

$$s := s_{k-1}, \quad q := q_k, \quad T := T_{k+1:i} := \prod_{j=k+1}^{i} q_j.$$

## B.2 SIGNAL-TO-NOISE RATIO

Define the SNR at horizon $i$ with respect to $\theta_k$ ($k \leq i$) as

$$\mathrm{SNR}_i(\theta_k) = \frac{\|\mathbb{E}\bar{g}_k\|^2}{\mathbb{E}\|\bar{g}_k - \mathbb{E}\bar{g}_k\|^2}.$$

To calculate this, we determine the mean and variance of $\bar{g}_k$.

**Mean.** For all $k \leq i$,

$$\mathbb{E}[\bar{g}_k] = g_k = \frac{s_i}{q_k}\nabla_{\theta^{(k)}}q_k = s\,T\,\nabla_{\theta^{(k)}}q_k. \tag{1}$$

**Variance identity (LOO).** Let $\mu := \mathbb{E}[R_i] = s\,q\,T$. The LOO variance decomposition gives

$$\mathrm{Var}(\bar{g}_k) = \frac{1}{N}\mathbb{E}\big[(R_i-\mu)^2 Z_k Z_k^\top\big] - \alpha_N\,g_k g_k^\top, \qquad \alpha_N = \frac{N-2}{N(N-1)}. \tag{2}$$

**Upstream $k < i$.** Conditioning on $I_{k-1} = 1$ and decoupling the tail as $\tilde{C} \sim \mathrm{Bernoulli}(T)$ independent of $B_k$, a direct enumeration yields

$$\mathbb{E}\big[(R_i-\mu)^2\,\|Z_k\|^2\big] = s\,\frac{T}{q(1-q)}\Big[(1-q)-2sTq+2sTq^2+s^2Tq^2\Big]\|\nabla q\|^2, \tag{3}$$

$$\Rightarrow \quad \mathbb{E}\big\|\bar{g}_k - \mathbb{E}\bar{g}_k\big\|^2 = \frac{s}{N}\frac{T}{q(1-q)}\Big[(1-q)-2sTq+2sTq^2+s^2Tq^2\Big]\|\nabla q\|^2 - \alpha_N\,s^2T^2\,\|\nabla q\|^2. \tag{4}$$

In particular, as $T \downarrow 0$,

$$\mathbb{E}\big\|\bar{g}_k - \mathbb{E}\bar{g}_k\big\|^2 = \frac{s\,T}{N}\frac{\|\nabla q\|^2}{q} + O\Big(\frac{s\,T^2}{N}\|\nabla q\|^2\Big) - \alpha_N s^2 T^2\|\nabla q\|^2. \tag{5}$$

**Frontier** $k = i$. Here $T \equiv 1$ and $R_i = I_{i-1} B_i$:

$$\mathbb{E}\big[(R_i - \mu)^2 \|Z_i\|^2\big] = s \left[\frac{(1 - sq)^2}{q} + \frac{s^2 q^2}{1 - q}\right] \|\nabla q\|^2, \tag{6}$$

$$\Rightarrow \quad \mathbb{E}\big\|\bar{g}_i - \mathbb{E}\bar{g}_i\big\|^2 = \frac{s}{N} \left[\frac{(1 - sq)^2}{q} + \frac{s^2 q^2}{1 - q}\right] \|\nabla q\|^2 - \alpha_N s^2 \|\nabla q\|^2. \tag{7}$$

Define

$$F(s, q, T) := (1 - q) - 2sTq + 2sTq^2 + s^2 T q^2 = (1 - q)(1 - 2sTq) + s^2 T q^2.$$

We obtain the exact formula

$$\mathrm{SNR}_i(\theta_k) = \frac{N \, s \, T \, q(1 - q)}{F(s, q, T)} \cdot \frac{1}{1 - \frac{N-2}{N-1} \cdot \frac{sT \, q(1-q)}{F(s,q,T)}}. \tag{8}$$

In the upstream regime with a small tail $T \ll 1$ (the typical long-horizon situation),

$$\mathrm{SNR}_i(\theta_k) = N \, s \, T \, q \cdot \left(1 + O\big(\tfrac{T}{1-q}\big)\right) \times \left(1 + O(sT)\right) = \Theta_{s,T}\big(N \, s \, T \, q\big) = \Theta(N \, s_i). \tag{9}$$

At the frontier $k = i$ one recovers the familiar form $\mathrm{SNR}_i(\theta_i) = \Theta_{s_{i-1}}(N \, s_{i-1} \, q_i (1 - q_i))$.

Therefore, when training at horizon $i$, the SNR for $\theta_k$ (with $k \le i$) scales in one of two ways:

$$\text{Upstream } (k < i): \quad \mathrm{SNR}_i(\theta_k) = \Theta(N \, s_i), \tag{10}$$

$$\text{Frontier } (k = i): \quad \mathrm{SNR}_i(\theta_i) = \Theta\big(N \, s_{i-1} \, q_i (1 - q_i)\big). \tag{11}$$

### B.3 PER-UPDATE IMPROVEMENT AND BATCH SIZE

Under the assumption that $s_i(\theta)$ is $L$-smooth, taking an update $\theta_k^+ = \theta_k + \eta \bar{g}_k$ (holding all $\theta_{j \neq k}$ fixed) results in expected improvement

$$\mathbb{E}\big[s_i(\theta^+) - s_i(\theta)\big] \ge \eta \left(\frac{s_i}{q_k}\right)^2 \|\nabla_{\theta_k} q_k\|^2 \Big(1 - \tfrac{L\eta}{2}\Big(1 + \tfrac{1}{\mathrm{SNR}_i(\theta_k)}\Big)\Big).$$

This follows from standard analysis of SGD on smooth functions (Duchi, 2018). It is maximised when $\eta = 1/(L(1 + 1/\mathrm{SNR}_i(\theta_k)))$, giving

$$\mathbb{E}\big[s_i(\theta^+) - s_i(\theta)\big] \ge \underbrace{\frac{s_{i-1}^2 \|\nabla_{\theta_k} g_k\|^2}{2 \, q_k^2 \, L}}_{:= \Delta_{i,k}^{(0)}} \cdot \frac{1}{1 + 1/\mathrm{SNR}_i(\theta_k)}. \tag{12}$$

We call $\Delta_{i,k}^{(0)}$ the noiseless improvement. To achieve a constant $\beta \in (0, 1)$ fraction of the noiseless gain requires

$$\mathrm{SNR}_i(\theta_k) \ge \frac{\beta}{1 - \beta}. \tag{13}$$

We now instantiate equation 8–equation 13 to compare training regimes in Appendix B.4.

### B.4 REGIMES AND CONSEQUENCES

**Single-step only (train only at $i = 1$).** When training only at $i = 1$, the required batch size and per-update noiseless improvement clearly do not depend on $H$. This method can only raise the atomic reliability $p$ (as $\sigma_1 = 1$), so even as $p \to 1$ long-horizon success remains bounded by $\prod_{j=2}^{H} \sigma_j$. However, given some target success probability $c \in (0, 1)$, increasing $p$ does increase the horizon $h$ at which $s_h \ge c$, with $h = \left(\ln c - \ln \prod_{j=2}^{h} \sigma_j\right) / \ln p$ given $c \le \prod_{j=2}^{h} \sigma_j$.

**Direct full horizon (train only at $i = H$).** If we train directly at horizon $H$, equation 10 gives for all $\theta_{k \neq 0}$ $\mathrm{SNR}_H(\theta_k) = \Theta(N \, s_H)$. Under the assumption that at initialization $q_j \in [\delta, 1 - \delta]$, $s_H = \Theta(e^{-H})$. We therefore have that

$$N = \Theta(e^H), \qquad \Delta_{H,k}^{(0)} = \Theta\left(e^{-2H} \|\nabla_{\theta^{(k)}} q_k\|^2\right).$$

The required batch size to achieve a constant fraction of the noiseless improvement is exponential, while the noiseless improvement decays exponentially, making training directly at large $H$ effectively impossible. In fact, direct training at horizon $H$ is worse than single-step training, as the signal is too small to effectively raise $p$ $\left(\mathrm{SNR}_H(\theta_0) = \Theta(N \, H \, e^{-H})\right)$.

**Curriculum over depths.** Given a target success probability $s_H \geq c \in (0, 1)$, we can ensure that curriculum training achieves this by only progressing the horizon $i$ when $q_i \geq 1 - \epsilon$, such that $(1 - \epsilon)^H \geq c$, and therefore $\epsilon \sim (-\ln c)/H$. If we assume that the earlier stages have been learned so $s_{i-1} \geq (1 - \epsilon)^{i-1} \geq c$, then equation 11 gives us that

$$N = \Theta\left(\frac{1}{\epsilon}\right) = \Theta(H), \qquad \Delta_{i,i}^{(0)} = \Theta\left(\|\nabla_{\theta^{(i)}} q_i\|^2\right).$$

$N$ depends on $\epsilon$ as $q_i \to 1 - \epsilon$, giving us a batch size that scales linearly with $H$. The noiseless improvement is independent of $H$ and $i$, and so under mild conditions on $\|\nabla_{\theta^{(i)}} q_i\|^2$, such that it shrinks at most polynomially in $H$ as $q_i \to 1 - \epsilon$, we achieve overall polynomial sample complexity for curriculum training.

**Uniform mixture over lengths.** Sample $I \sim \mathrm{Unif}\{1{:}H\}$ and run the depth-$I$ estimator; for a fixed block $i$, the per-*iteration* SNR obtained for its update averages to

$$\mathbb{E}_I[\mathrm{SNR}_i] = \Theta\left(\frac{N}{H} \, s_{i-1} \, q_i \sum_{t=0}^{H-i} T_{i+1:i+t}\right), \qquad T_{a:b} := \prod_{\ell=a}^{b} q_\ell \; (T_{a:a-1} := 1).$$

*Frontier phase.* We say horizon $i$ is at the frontier when earlier skills are sufficiently reliable while deeper ones are not yet learnt, namely

$$s_{i-1} \geq c \quad \text{for some fixed } c \in (0, 1) \qquad \text{and} \qquad \sum_{t=0}^{H-i} T_{i+1:i+t} = \Theta(1).$$

During this frontier phase,

$$\mathbb{E}_I[\mathrm{SNR}_i] = \Theta\left(\frac{N}{H} \, s_{i-1} \, q_i\right).$$

Whenever we sample a batch with $I = i$, we obtain the same noiseless improvement and batch size scaling as curriculum training, with

$$N = \Theta\left(\frac{1}{\epsilon}\right) = \Theta(H), \qquad \Delta_{i,i}^{(0)} = \Theta\left(\|\nabla_{\theta^{(i)}} q_i\|^2\right).$$

Whenever we sample $I \neq i$, we see negligible change as samples with $h < i$ cannot improve $q_i$, and samples with $h > i$ have per-iteration gain that scales with $s_{h-1}^3$.

$$\mathbb{E}_I[\Delta_h \text{ per iter}] = \Theta\left(\frac{N}{H} \, s_{h-1}^3 \, \|\nabla_{\theta_i} q_h\|^2\right).$$

Therefore, it takes $\sim H$ times longer to train with uniform sampling than with curriculum, due to only a fraction $1/H$ of the updates being "useful" at each frontier $i \in \{1, \cdots, H\}$.

## B.5 DENSE REWARDS (REWARD-TO-GO WITH STATE-VALUE BASELINE)

We replace the terminal-only objective at depth $i$ with dense stepwise rewards $r_t := \mathbf{1}\{I_{t-1}B_t = 1\}$ and train with reward-to-go at horizon $i$. Fix a block $k \le i$. Define

$$\Sigma_{k,i} := \sum_{t=k}^{i} \prod_{j=k+1}^{t} B_j, \qquad S_{k,i} := \mathbb{E}[\Sigma_{k,i}] = \sum_{d=0}^{i-k} T_{k+1:k+d}, \qquad S_{k,i}^{(2)} := \mathbb{E}[\Sigma_{k,i}^2] = \sum_{d=0}^{i-k} (2d+1)\, T_{k+1:k+d},$$

with $T_{a:b} := \prod_{\ell=a}^{b} q_\ell$ and $T_{a:a-1} := 1$. Let $s := s_{k-1}$ and $q := q_k$. The per-sample score is

$$Z_k = I_{k-1} \frac{B_k - q}{q(1-q)} \nabla_{\theta^{(k)}} q_k,$$

and we use the *state-value* (per-sample, action-independent) baseline

$$b = \mathbb{E}[R \mid I_{k-1}, \Sigma_{k,i}] = I_{k-1}\, q\, \Sigma_{k,i},$$

so each summand is $X := (R-b)Z_k$ and $\bar{g}_k = \frac{1}{N} \sum_{g=1}^{N} X_g$ with i.i.d. terms.

**Mean (signal).** Since $(R-b) = I_{k-1}\Sigma_{k,i}(B_k - q)$ and $Z_k = I_{k-1}\frac{(B_k-q)}{q(1-q)}\nabla q$,

$$\mathbb{E}[\bar{g}_k] = \mathbb{E}[X] = \mathbb{E}\left[I_{k-1}\Sigma_{k,i}\frac{(B_k-q)^2}{q(1-q)}\right]\nabla q = s\, S_{k,i}\, \nabla_{\theta^{(k)}} q_k.$$

**Variance decomposition and exact MSE.** Because we use a per-sample baseline, there is no LOO cross-term and

$$\mathrm{Var}(\bar{g}_k) = \frac{1}{N}\mathrm{Var}(X) = \frac{1}{N}\Big(\mathbb{E}\big[(R-b)^2\|Z_k\|^2\big] - \|\mathbb{E}[(R-b)Z_k]\|^2\Big).$$

A one-step Bernoulli calculation yields

$$\mathbb{E}\big[(R-b)^2\|Z_k\|^2\big] = s\, S_{k,i}^{(2)}\frac{1-3q(1-q)}{q(1-q)}\|\nabla_{\theta^{(k)}} q_k\|^2, \qquad \|\mathbb{E}[(R-b)Z_k]\|^2 = s^2 S_{k,i}^2 \|\nabla_{\theta^{(k)}} q_k\|^2.$$

Hence the exact mean-squared error (MSE) is

$$\mathbb{E}\big\|\bar{g}_k - \mathbb{E}\bar{g}_k\big\|^2 = \frac{s}{N}\left(\frac{1-3q(1-q)}{q(1-q)} S_{k,i}^{(2)} - s\, S_{k,i}^2\right)\|\nabla_{\theta^{(k)}} q_k\|^2. \tag{14}$$

**SNR** Using $\|\mathbb{E}\bar{g}_k\|^2 = s^2 S_{k,i}^2 \|\nabla q\|^2$ and equation 14,

$$\mathrm{SNR}_i(\theta_k) = \frac{N\, s\, S_{k,i}^2}{\frac{1-3q(1-q)}{q(1-q)} S_{k,i}^{(2)} - s\, S_{k,i}^2}. \tag{15}$$

Since $1-3q(1-q) \in [1/4, 1]$, we obtain the MSE bounds

$$\frac{s}{N}\Big(\frac{S_{k,i}^{(2)}}{4\, q(1-q)} - s S_{k,i}^2\Big)\|\nabla q\|^2 \le \mathbb{E}\|\bar{g}_k - \mathbb{E}\bar{g}_k\|^2 \le \frac{s}{N} \cdot \frac{S_{k,i}^{(2)}}{q(1-q)}\|\nabla q\|^2,$$

and therefore (whenever the positive term dominates the $s S_{k,i}^2$ subtraction, e.g. away from extremely large $s$)

$$\mathrm{SNR}_i(\theta_k) = \Theta\left(N\, s\, q(1-q) \cdot \frac{S_{k,i}^2}{S_{k,i}^{(2)}}\right).$$

**Tail regularity and equivalence with curriculum.** If the tail reliabilities are bounded away from the boundary, $q_{k+1}, \ldots, q_i \in [\delta, 1 - \delta]$ for some $\delta \in (0, \frac{1}{2}]$, then

$$S_{k,i}^{(2)} = \Theta\big(S_{k,i}^2\big) \quad \text{(constants depend only on } \delta\text{),}$$

yielding

$$\mathrm{SNR}_i(\theta_k) = \Theta\big(N\, s_{k-1}\, q_k (1 - q_k)\big).$$

This *removes* the tail-reach penalty $T_{k+1:i}$ that appears with terminal-only rewards and exactly matches the curriculum-frontier scaling at stage $i$ (where curriculum also yields $\Theta(N\, s_{i-1} q_i (1 - q_i))$). Consequently, the batch size needed to attain a $\beta$-fraction of the noiseless improvement in equation 12 is the same order as under curriculum:

$$N_i = \Theta\left(\frac{1}{s_{i-1}\, q_i (1 - q_i)} \cdot \frac{\beta}{1 - \beta}\right),$$

and with the standard curriculum gate $q_i \geq 1 - \epsilon$ and $s_{i-1} \geq c$ (so $\epsilon \sim (-\ln c)/H$), this is $N_i = \Theta\big(\frac{H}{c(-\ln c)} \cdot \frac{\beta}{1-\beta}\big)$.

## C    DECREASING SAMPLE COMPLEXITY FOR LONGER TRAINING DATA

In Section 6, we train with the following sample count settings skewed towards shorter samples.

- Baseline: L1 100, L2 100, L3 100, L4 100

- Setting 1: L1 150, L2 115, L3 85, L4 50

- Setting 2: L1 175, L2 100, L3 75, L4 50

- Setting 3: L1 200, L2 125, L3 50, L4 25

And we concluded that we can recover the baseline performance with a skewed trianing distribution, as long as well spend more compute. However, this is a small search space.

To support section 6 better, we scale up the search space by simplifying our experimental setting. In particular, we consider training a 135M-parameter model on integer multiplication problems through SFT. We generate the multiplication problems by sampling two operands, and writing out the chain of computations. We define **length** as the sum of number of digits of both operands, analogous to the number of chained GSM problems in our primary setting. Then we can separate the training dataset into bins grouped by distinct lengths. We vary the length distribution of training dataset by varying the samples in each length bins. Finally we associate a **cost** to each data length, which represents the cost of generating the data. This metric mirrors the real-world concern that longer data is harder to collect. For multiplication, the cost of each length is equal to the length.

## C.1    Trade-off between data cost and compute

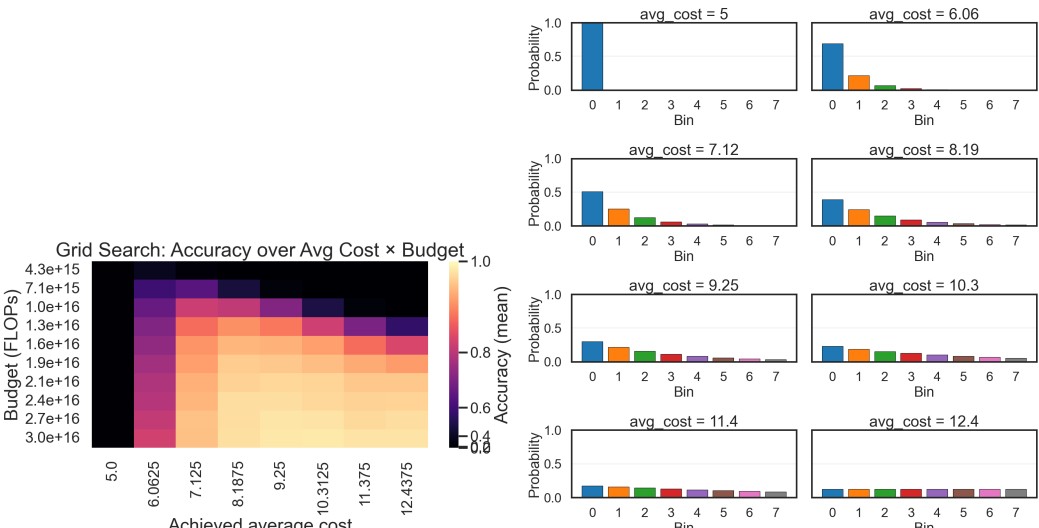

Figure 7: Right: We parameterize different training distribution using a single cost scalar. Left: Sweeping different choices of training budget and distribution costs

Using the multiplication task, we study the trade-off between (1) skewedness towards shorter lengths in the training distribution and (2) total training budget. Figure 7 sweeps over many choices of budget and cost. A first observation is that for the same target accuracy, a training run can either have a lower cost data distribution and use more budget, or vise versa. Figure 7 also shows other relationships. For example, the rate of learning seems to slowly decrease as we increase cost. We believe this is because as we shift more weight to longer examples, the multiplication becomes harder to learn overall.

## C.2    Searching for the minimum cost distribution under the same budget

By principally testing different data mixtures we show that longer horizon training requires less samples. Here, we create 3 length bins, which evenly divides the data length in the dataset. We keep the training budget the same, and vary the data mixture of the length bins to find the mixture distribution with the least average cost but still keeps high performance after training. We visualize this search procedure in Figure 8, which shows a distinct feasible region for the 3-bin probability distribution where training runs are successful.

## D    Additional Experimental Details.

We trained the Qwen 2.5 3B Instruct model using the hyperparameters outlined in Table 5. The training was conducted for 200 optimization steps for each problem length in our curriculum, where each step processed a single sample, for a total of 200 samples per horizon. We utilized the Group-Relative Policy Optimization (GRPO) training objective. For each problem length, we evaluated the model every 50 steps and selected the best checkpoint based on validation performance. This checkpoint was then used as the initialization for training on the subsequent, longer-horizon problem length.

The maximum completion length was dynamically adjusted based on the number of sub-problems to accommodate the increasing reasoning horizon. Specifically, we used a maximum completion length of 768 for 1-subproblem tasks, 1024 for 2-subproblem tasks, 1280 for 3-subproblem tasks, and 1536 for both 4- and 5-subproblem tasks. Dynamically increasing the completion length was an important factor to achieve good performance, as it allowed the model enough token space to solve the problem while also constraining it to the minimum length required to complete the task.

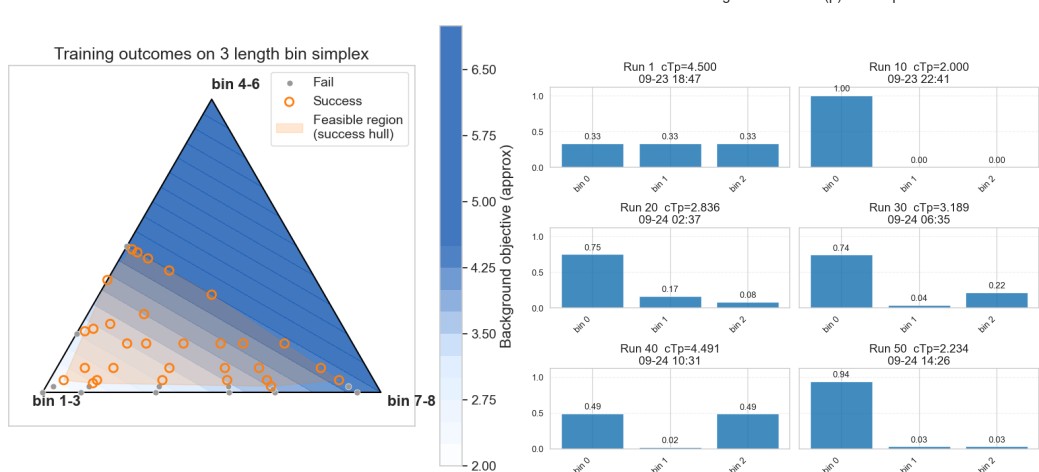

Figure 8: Left: Training trails with 3 length bin data distribution, plotted over the probability simplex. The blue gradient is the "cost" of the distribution, as defined in C.1. Each dot on the simplex is a training run with the specific data distribution. We start from the uniform distribution (middle point) and send 8 rays to the cheaper half of the simplex boundary (we did not explore the more expensive half). Then we bisect each rays to find the feasibility boundary along this ray. Overall, there is a convex feasible region that forms close to the simplex boundaries, and we are able to find data distributions much cheaper.Right: Examples of different 3-bin training distributions during the search.

| Parameter | Qwen 2.5 3B Instruct | Qwen 2.5 7B Instruct | Llama 3.2 3B Instruct |
|---|---|---|---|
| Training Steps per Horizon | 200 | 200 | 200 |
| Samples per Horizon | 200 | 200 | 200 |
| Number of Generations per Prompt | 16 | 16 | 16 |
| Learning Rate | $5 \times 10^{-6}$ | $2 \times 10^{-6}$ | $5 \times 10^{-6}$ |
| Learning Rate Scheduler | Cosine | Cosine | Cosine |
| Warmup Steps | 30 | 30 | 20 |
| Max Gradient Norm | 0.1 | 0.1 | 0.1 |
| Loss Type | Dr. GRPO | Dr. GRPO | Dr. GRPO |

Table 5: Hyperparameters used for the curriculum-based RL training stages across different models.

We implemented GRPO training using the Verifiers library(Brown, 2025).

---

**System prompt for RL training**

Respond in the following format, with only the numerical answer between the <answer> tags:
<reasoning>
...
</reasoning>
<answer>
...
</answer>

---

**Example of chained GSM8K problem**

**Step 1: Solve the following math problem step by step:**

---

Ralph watches TV for 4 hours a day from Monday to Friday, and 6 hours a day on Saturday and Sunday. How many hours does Ralph spend watching TV in one week?

**Step 2: Take your final answer from Step 1 and substitute it for Z in the following problem:**

Sonny received Z boxes of cookies from his friend yesterday. He gave 12 to his brother, 9 to his sister, and he gave 7 to his cousin. How many boxes of cookies were left for him?

**Write out the updated version of the problem with the number from Step 1 in place of Z.**

**Step 3: Solve the updated problem from Step 2 step by step.**

**Step 4: Take your final answer from Step 3 and substitute it for U in the following problem:**

In a restaurant, the chef prepared 17 meals to sell for lunch. During lunch, he sold 12 meals. After lunch, the chef prepared another U meals for dinner. How many meals does the chef have for dinner, assuming he can use the remaining lunch meals as well?

**Write out the updated version of the problem with the number from Step 3 in place of U.**

**Step 5: Solve the updated problem from Step 4 step by step. In the end, provide only the final numerical answer.**

——

**Answer: 9**

The same hyperparameters were applied to our baselines in Table 1. For these baselines, we trained up to an equal amount of compute as our main experiments and selected the best checkpoint from each run based on validation performance.

To evaluate our model's generalization capabilities, we performed a series of zero-shot evaluations (sampling temperature 0.1) on a variety of benchmarks, including AIME 2024, AIME 2025, MMLU Pro Math (Wang et al., 2024), GSM Symbolic (Mirzadeh et al., 2025), MATH-500 (Hendrycks et al., 2021), LongBench-v2(Bai et al., 2025), Hash-hop(Magic, 2024), and GPQA(Rein et al., 2024) (Tables 2, 6). For the Hash-hop benchmark, we computed the average accuracy across multiple settings, including context lengths of 10k, 20k, and 30k characters, and 1, 2, 3, and 4 hops.

Besides harder math benchmarks, our curriculum-based training generalizes to benchmarks that require long-context and complex reasoning, even though our models were only trained on composed mathematical problems. The results on LongBench-v2, Hash-hop, and GPQA show a consistent improvement in performance as the training horizon increases, demonstrating that our method imparts transferable skills such as state-tracking and the ability to reason over long sequences. For example, performance on LongBench-v2 increases from 35.00% (untrained) to 37.90% after training up to a 5-subproblem horizon.

The robustness and generality of our method were demonstrated by applying it to two additional language models: Qwen 2.5 7B Instruct and Llama 3.2 3B Instruct. The training parameters for all models are detailed in Table 5.

For the Qwen 2.5 7B Instruct model, we constructed chained problems from a more challenging source, the MATH dataset. We connected subproblems with integer-valued answers by applying simple operations (e.g., addition or subtraction) to generate the numerical input for the next problem. The training showed a strong, consistent performance lift, with mean accuracy on multi-step problems rising from 45.50% to 50.65% (Table 7). This improvement transferred to out-of-domain benchmarks, validating the method's ability to generalize beyond the specific training domain.

| | Generalization to Long Context Benchmarks | | |
|---|---|---|---|
| Model / setting | LongBench-v2 | Hash-hop | GPQA |
| **Instruct model** | 35.00 | 15.98 | 25.00 |
| *Standard RLVR on GSM8K* | | | |
| **GSM8K RLVR** | 35.30 | 14.76 | 26.56 |
| *Curriculum RL on Composed GSM8K Problems* | | | |
| **Len-2 GSM8K** | 36.20 | 16.17 | 25.22 |
| **Len-3 GSM8K** | 37.10 | 17.62 | 26.12 |
| **Len-4 GSM8K** | 36.20 | **18.98** | 26.34 |
| **Len-5 GSM8K** | **37.90** | 18.73 | **27.23** |
| | (+8.3%) | (+17.4%) | (+8.9%) |

Table 6: Performance on long context benchmarks improves significantly with GSM8K RL curriculum training stages. Training on increasing complexity of GSM8K leads to strong out-of-domain generalization.

| | Accuracy on MATH Problems | | | | | Accuracy on Harder Problems | |
|---|---|---|---|---|---|---|---|
| Model / setting | Len-1 | Len-2 | Len-3 | Len-4 | Mean | Symbolic P2 | LongBench v2 |
| **Instruct model** | 74.00 | 52.60 | 29.40 | 26.00 | 45.50 | 61.36 | 33.60 |
| *Standard RLVR on MATH* | | | | | | | |
| **MATH RLVR** | 76.20 | 53.80 | 32.00 | 29.00 | 47.75 | 64.96 | 34.50 |
| *Curriculum RL on Composed MATH Problems* | | | | | | | |
| **Len-2 MATH** | **77.00** | 56.20 | 34.40 | 27.66 | 48.82 | **65.60** | 34.50 |
| **Len-3 MATH** | 76.20 | 56.00 | 35.40 | 28.86 | 49.12 | 65.32 | 34.50 |
| **Len-4 MATH** | 76.80 | **56.60** | **37.80** | **31.40** | **50.65** | 64.88 | **35.30** |
| | (+4.1%) | (+7.6%) | (+28.6%) | (+20.8%) | (+11.3%) | (+6.9%) | (+5.1%) |

Table 7: Long Horizon MATH Training on **Qwen 2.5 7B Instruct**. Curriculum stages lead to significant improvements in in-domain performance and generalization metrics. Len-1 refers to MATH-500 dataset.

For the Llama 3.2 3B Instruct model, we used the same chained GSM8K problems as our primary experiments. The results for this model are presented in Table 8. The successful application of our method to a different model family demonstrates its effectiveness across diverse architectural designs and confirms that our curriculum learning framework is a robust and generalizable method for improving long-horizon reasoning.

| | Accuracy on GSM8K Problems | | | | | Accuracy on Harder Problems | |
|---|---|---|---|---|---|---|---|
| Model / setting | Len-1 | Len-2 | Len-3 | Len-4 | Mean | Symbolic P1 | AIME Mean |
| **Instruct model** | 78.00 | 11.83 | 4.42 | 1.34 | 23.90 | 54.84 | 2.09 |
| *Curriculum RL on Composed GSM8K Problems* | | | | | | | |
| **Len-1 GSM8K** | 79.00 | 13.28 | 7.14 | 2.14 | 25.39 | 55.16 | 2.87 |
| **Len-2 GSM8K** | 80.20 | 35.06 | 15.99 | 6.43 | 34.42 | 57.52 | 2.92 |
| **Len-3 GSM8K** | **80.60** | **35.27** | **17.35** | **6.70** | **34.98** | **57.75** | **3.18** |
| | (+3.3%) | (+198.1%) | (+292.5%) | (+400.0%) | (+46.4%) | (+5.3%) | (+52.2%) |

Table 8: Long Horizon GSM8K Training on **Llama 3.2 3B Instruct**. Curriculum stages lead to significant improvements in in-domain performance and generalization metrics.

# E  QUALITATIVE EXAMPLE

---

**LHR Trained Qwen-2.5-3B on Explicit-Horizon Problems**

**Long-Horizon Question**

Solve the following math problem step by step:

Step 1: Janet's ducks lay 16 eggs per day. She eats three for breakfast every morning and bakes muffins for her friends every day with four. She sells the remainder at the farmers' market daily for $2 per fresh duck egg. How much in dollars does she make every day at the farmers' market?

Step 2: Take your final answer from Step 1 and substitute it for W in the following problem: Ella earned $W for 3 hours of dog walking. If she continues to walk dogs at the same rate, how much would she earn after 12 hours?
Write out the updated version of the problem with the number from Step 1 in place of W.

Step 3: Solve the updated problem from Step 2 step by step.

Step 4: Take your final answer from Step 3 and substitute it for U in the following problem: Peter purchased U popsicles at $0.25 each. He also purchased 4 ice cream bars at $0.50 each. How much did he pay in total in dollars?
Write out the updated version of the problem with the number from Step 3 in place of U.

Step 5: Solve the updated problem from Step 4 step by step.

Step 6: Take your final answer from Step 5 and substitute it for Y in the following problem: Fred was preparing for a party to be held in four days. So, he made Y gallons of root beer on the first day and put them in the refrigerator cooler. But later that evening, his children discovered the delicious nectar and robbed the cooler, drinking 4 of those gallons of root beer. On the second day, his wife Barbie also discovered the root beer and accidentally spilled 7 gallons. On the third day, Fred's friend Ronnie visited Fred's house and helped himself to the root beer, further reducing the amount remaining by 5 gallons. On the fourth day, 3 people showed up for the party. If Fred and the others shared the remaining root beer equally, how much was available for each to drink during the party?
Write out the updated version of the problem with the number from Step 5 in place of Y.

Step 7: Solve the updated problem from Step 6 step by step.

Step 8: Take your final answer from Step 7 and substitute it for Z in the following problem: Craig has 2 twenty dollar bills. He buys six squirt guns for $2 each. He also buys Z packs of water balloons for $3 each. How much money does he have left?
Write out the updated version of the problem with the number from Step 7 in place of Z.

Step 9: Solve the updated problem from Step 8 step by step. In the end, provide only the final numerical answer.

**Ground truth answer:** 25

**Untrained model response**

<reasoning>

Janet's ducks lay 16 eggs per day. She eats 3 eggs for breakfast and uses 4 eggs to make muffins. The remaining eggs are sold at the farmers' market. The number of eggs sold is calculated as follows:

$$16 - 3 - 4 = 9$$

Since she sells each egg for $2, her daily earnings at the farmers' market are:

$$9 \times 2 = 18$$

So, Janet makes $18 every day at the farmers' market. This value is substituted for W in the next problem.

Ella earned $18 for 3 hours of dog walking. If she continues to walk dogs at the same rate, her hourly rate is:

$$\frac{18}{3} = 6$$

Ella's hourly rate is $6. This value is substituted for W in the updated problem.

---

Ella earns $6 per hour. If she continues to walk dogs at the same rate, how much would she earn after 12 hours?

$$6 \times 12 = 72$$

So, Ella would earn $72 after 12 hours. This value is substituted for U in the next problem.

Peter purchased $72 at $0.25 each. He also purchased 4 ice cream bars at $0.50 each. The total cost is calculated as follows:

$$72 \times 0.25 + 4 \times 0.50 = 18 + 2 = 20$$

So, Peter paid $20 in total. This value is substituted for U in the next problem.

Peter paid $20. If he purchased 20 packs of root beer, how much was available for each to drink during the party?

$$\frac{20}{4} = 5$$

So, 5 gallons of root beer were available for each to drink during the party. This value is substituted for Y in the next problem.

Fred made 5 gallons of root beer. On the second day, his wife spilled 7 gallons, on the third day, his friend Ronnie helped himself to 5 gallons, and on the fourth day, 3 people showed up for the party. The remaining root beer is calculated as follows:

$$5 - 7 - 5 - 3 = -10$$

Since the remaining amount is negative, it means Fred ran out of root beer and did not have any left for the party. This value is substituted for Y in the next problem.

Fred did not have any root beer left. If he purchased 0 gallons of root beer, how much money does he have left after buying 2 twenty dollar bills?

$$2 \times 20 = 40$$

So, Fred spent $40 on the 2 twenty dollar bills. This value is substituted for Y in the next problem.

Craig has $40. He buys 6 squirt guns for $2 each and 3 packs of water balloons for $3 each. The total cost is calculated as follows:

$$6 \times 2 + 3 \times 3 = 12 + 9 = 21$$

So, Craig spent $21 on the squirt guns and water balloons. The amount of money he has left is:

$$40 - 21 = 19$$

So, Craig has $19 left. This value is substituted for Z in the next problem.

Craig has $19 left. If he buys Z packs of water balloons for $3 each, how much money does he have left?

$$19 - 3 \times 3 = 19 - 9 = 10$$

So, Craig has $10 left. This value is substituted for Z in the next problem.

Craig has $10 left. If he buys 10 packs of water balloons for $3 each, how much money does he have left?

$$10 - 10 \times 3 = 10 - 30 = -20$$

Since the remaining amount is negative, it means Craig does not have enough money to buy 10 packs of water balloons. The final numerical answer is -20.
</reasoning>
<answer>

-20
</answer>

### Analysis (Errors in the untrained model's response)
**State-tracking errors**

1. **Step 6→7 (after "Peter paid $20"):** Took shortcut $20/4 = 5$ and then *reused* $Y = 5$ as the starting gallons for Fred ("Fred made 5 gallons..."). Correct carry-over for Step 6 is $Y = 20$.

2. **Step 8→9 (Craig stage):** From "Craig has $19 left." it set $Z = 19$ but computed $19 - 3 \times 3$ instead of $19 - 3Z$. Next line used $Z = 10$, showing inconsistent substitution.

**Problem-logic errors**

1. **Step 6 / start of Step 7:** Treated $20 as gallons and skipped events, doing $20/4$ instead of $20 - 4 - 7 - 5 = 4$ then $4/4 = 1$.

2. **Step 7:** Subtracted people as gallons, computing $5 - 7 - 5 - 3$ instead of dividing the remaining gallons among $4$ people.

### LHR model response
<reasoning>

1. First, we need to calculate how many eggs Janet sells each day.
   - Janet's ducks lay 16 eggs per day.
   - She eats 3 eggs for breakfast.
   - She bakes muffins with 4 eggs.
   So, the number of eggs she sells each day is:
   $$16 - 3 - 4 = 9 \text{ eggs}$$

   Janet sells the eggs at $2 per egg. Therefore, the amount she makes every day at the farmers' market is:
   $$9 \times 2 = \$18$$

2. Substitute the final answer from Step 1 (18) into the problem:
   Ella earned $18 for 3 hours of dog walking. If she continues to walk dogs at the same rate, how much would she earn after 12 hours?

3. Ella's rate of earning is $18 for 3 hours. Therefore, her hourly rate is:
   $$\frac{18}{3} = \$6 \text{ per hour}$$

   To find out how much she would earn after 12 hours, we calculate:
   $$6 \times 12 = \$72$$

4. Substitute the final answer from Step 3 (72) into the problem:
   Peter purchased 72 popsicles at $0.25 each. He also purchased 4 ice cream bars at $0.50 each. How much did he pay in total in dollars?

5. The cost of the popsicles is:
   $$72 \times 0.25 = \$18$$

   The cost of the ice cream bars is:
   $$4 \times 0.50 = \$2$$

   The total cost is:
   $$18 + 2 = \$20$$

6. Substitute the final answer from Step 5 (20) into the problem:
Fred was preparing for a party to be held in four days. So, he made 20 gallons of root beer on the first day and put them in the refrigerator cooler. But later that evening, his children discovered the delicious nectar and robbed the cooler, drinking 4 of those gallons of root beer. On the second day, his wife Barbie also discovered the root beer and accidentally spilled 7 gallons. On the third day, Fred's friend Ronnie visited Fred's house and helped himself to the root beer, further reducing the amount remaining by 5 gallons. On the fourth day, 3 people showed up for the party. If Fred and the others shared the remaining root beer equally, how much was available for each to drink during the party?

7. The amount of root beer remaining after each event is:
$$20 - 4 - 7 - 5 = 4 \text{ gallons}$$

Fred and 3 others (including himself) share the remaining 4 gallons equally. Therefore, each person gets:
$$\frac{4}{4} = 1 \text{ gallon}$$

8. Substitute the final answer from Step 7 (1) into the problem:
Craig has 2 twenty dollar bills. He buys six squirt guns for $2 each. He also buys Z packs of water balloons for $3 each. How much money does he have left?

9. The cost of the squirt guns is:
$$6 \times 2 = \$12$$

The cost of the water balloons is:
$$Z \times 3$$

The total cost is:
$$12 + 3Z$$

Craig has 2 * 20 = $40. After spending, the remaining money is:
$$40 - (12 + 3Z)$$

Substitute the final answer from Step 7 (1) for Z:
$$40 - (12 + 3 \times 1) = 40 - 15 = \$25$$

Final numerical answer: 25
</reasoning>
<answer>
25
</answer>

