# OpenReview forum: "h1: Bootstrapping Models to Reason over Longer Horizons via Reinforcement Learning"
_ICLR.cc/2026/Conference — Submitted to ICLR 2026_

### Official Review · Reviewer_WYvB · 2025-10-27

**Soundness:** 1
**Presentation:** 3
**Contribution:** 2
**Rating:** 4
**Confidence:** 3

**Summary:**

This paper introduces a new technique for doing synthetic task augmentation and curriculum generation for RL. The proposed augmentation method "h1" simply composes multiple short-horizon problems by using the output of previous problems as variables for future ones. The empirical evaluation is focused on a Qwen2.5 3B model trained on a composable version of GSM8K and evaluated across other math domains.

**Strengths:**

- Synthetic data augmentation for long-horizon reasoning is an under-explored, but very relevant research direction, as we want to scale RL to increasingly complex domains.
- Overall, the paper is well written and reads smoothly. I found the provided content and method description clear and intuitive.
- I found the cost analysis result in Section 6 insightful, providing interesting results on the tradeoff between cost and performance for curriculum learning, something I have not seen in prior work.

**Weaknesses:**

My **main concerns** about this work are that I found a large divide between the main claims made, the methodology, and the actual empirical results:
1) The paper makes several resounding claims, e.g., that their method "provides scalable synthetic long-horizon data, with explicit control over the horizon length and complexity without the need for new annotations." (lines 78-80). Yet even in the GSM8K settings, the proposed method requires annotating where variables of future claims can be augmented with previous answers, which is something extraneous to traditional RL datasets. Moreover, it is unclear to me how h1 would be applicable to domains beyond ones with numeric answers (e.g., game playing, coding, or scientific discovery, which the authors themselves mentioned in lines 38-39).
2) The current main results focus on a 3B Qwen 2.5 model on the authors' own synthetic version of GSM8K for multi-hop problems with limited output context. If I am interpreting Table 1 correctly, gains from h1 on this task appear only on the synthetic examples, with performance seemingly lower than regular compute matched non-augmented training L1 on the original set of problems (column 1 of Table 1). While the performance improvements on these longer synthetic problems are interesting, I do not think this result alone provides evidence for generalized "substantial performance gains on multi-step reasoning tasks" (lines 475-476), which I think would require testing the h1 model on other explicitly multi-hop problems across different domains after training.
3) When evaluating on a broader set of tasks in Table 2, the only baseline reported is the author's own RLVR implementation without *compute matching*. I am not sure why the other "compute matched" baselines from Table 1 are not also reported in Table 2.  Also, for the 7b Qwen experiments, I could not find the results on AIME@16. Given the 7B model has been widely used by prior reasoning works this would allow for comparison with a wider range of baselines. I would suggest expanding the range of evaluation tasks to include Minerva Math, OlympiadBench, and AMC as well, matching the evaluation setting of Dr.GRPO, and including this method as an independently trained baseline. Since Dr.GRPO itself is the method used by the authors for optimization, I think adding it as an independent baseline would allow readers to put into better perspective the actual impact of the methodology. At the moment, I do not think that claims like "while improvements obtained from RLVR on standard data is bounded by the base model’s capabilities, our method performs significantly better" (lines 76-78)or "Furthermore, our results show that the model learns genuinely new reasoning capabilities, rather than just refining existing ones." (lines 478-479) are justified, given that all absolute performance score reported with the 3B model for the reasoning tasks (e.g., AIME, MATH) are quite low and well below prior reasoning work.


**Other concerns**:
1) In Algorithm 1, it is described that at each training stage, the model's previous solution $y_h$ is used to construct future prompts, prepended in the dataset, for later curriculum iterations (Algorithm line 8). Thus, it is not clear to me how the "uniform mix" and "only long" baselines' data is constructed. Is this using ground-truth solutions perhaps? I could not find this information in the main text. Additionally, it is not entirely clear to me how compute was "equated" for these baselines. Since the main algorithm was run for 200 iterations at each stage (up to 5 stages), for how many iterations were the other baselines run for?

2) Qwen 2.5 3B instruct has not been post-trained with RL (unlike instruct models from the Qwen 3 family). Models from the Qwen 2.5 family have been shown to improve significantly with random rewards, and have often been tied to data leakage in the base model (e.g., [1]). To this end, I do not think that the sentence "Improving an Instruct model with RL is generally considered more difficult and gains signify performance improvements beyond just instruction tuning, which cannot be directly inferred for improvements on base models" (ln 250 253) is an accurate depiction of current findings.


3) A key component emerging in reasoning models is their ability to self-verify, correct, and re-attempt problems at any point of their reasoning chain [2]. The analysis in Appendix C, initiated in Section 3 seems to ignore this (e.g., the Equation in line 185, assuming intermediate findings are fixed) and makes other assumptions not applicable to the "single-turn" evaluation tasks chosen, that I do not think justify some of the strong definitive statements in the paper (e.g., from the abstract "Theoretically, we show that curriculum-based RL with outcome rewards achieves an exponential improvement in sample complexity over full-horizon training"). I would recommend toning down these claims (e.g., achieves -> "could achieve") and explicitly mentioning that these are made under several simplifying assumptions.

4) Looking at Table 4 in Appendix E, it seems the optimization was run with a quite low learning rate and gradient clipping magnitude. Were these choices due to a hyperparameter sweep and/or some instability observations? I could also not find this information in the paper.

5) In Appendix E, the authors also mention that they evaluated their models every 50 steps for each training horizon and selected the checkpoint with the highest performance on a separate validation set. I found this quite unusual, as reasoning RL methods train on less than a single epoch anyway, meaning validation and training performance should match, and report results with their final checkpoint. I think it would be important to clarify exactly what validation data is being used and whether it is from a different distribution than the training set. Additionally, I think validation curves and the collected performances every 50 steps for both the h1 model and all baselines should be added to the text, as this information is currently missing.

[1] Shao, Rulin, et al. "Spurious rewards: Rethinking training signals in rlvr." arXiv preprint arXiv:2506.10947 (2025).

[2] Guo, Daya, et al. "Deepseek-r1: Incentivizing reasoning capability in llms via reinforcement learning." arXiv preprint arXiv:2501.12948 (2025).

**Questions:**

I would appreciate it if the authors could address the areas of improvement and questions raised in the  Weaknesses section of my review, and clarify any potential misconceptions for the discussion phase.

---

> ### Author Response · Authors · 2025-11-25
>
> Dear Reviewer,
>
> Thank you for your time and valuable feedback! We appreciate the opportunity to address the points raised and further improve our work.
>
> 1. Our method only relies on having ground truth single-step labels, which is the structure that most datasets currently have. Given these ground truth labels, we can compose problems. All that is required is that the answers are in some way verifiable (not necessarily numeric) which would then allow further problems to depend on previous solutions. This dependence can be at two levels: (1) transformation chaining - where unit conversions can be made to the output of one question so it matches the input of the next one or (2) recompute chaining where subsequent problems can be modified to depend on previous solutions. By no new annotations, we mean no new data or human-annotations. Our chaining does not require new human annotations and we have now specified this clearly in the paper. While our training effort is based on data with numerical solutions, our method can work with code or other verifiable domains where previous solutions can be used as inputs or subroutines for new problems to create a multi-step dependency chain.
>
> 2. Gains from h1 appear on long-horizon examples from our test set, much longer synthetic examples (lengths 6, 7, and 8), and on several other benchmarks (MATH-500, AIME, MMLU Pro, GSM Symbolic P1, GSM Symbolic P2, Hash-hop, and Longbench-v2). Performance on just length 1 is within 1% when training using a long-horizon versus equal-compute only-L1 data, and there is not much reason to see substantial improvements beyond training on just length 1 when testing on the same horizon. Our performance does improve on all the other multi-step mathematical tasks stated here (beyond both an Instruct model baseline and length-1 RL training). However, to further support our claim of general reasoning performance improvements, we test on ReasoningGym [1] domains. ReasoningGym consists of a diverse set of (new) reasoning environments that allow us to evaluate crossdomain transfer and skill generalization. We test performance on propositional logic, graphs, algorithmic (sentence reordering), and algorithmic (matrix operations) and see substantial transfer beyond math and long-context domains. Solving these problems requires maintaining a working memory, graph traversal, rule following, and general multi-hop reasoning capabilities.
> | Model / setting | Propositional logic | Graphs (largest island) | Algorithmic (sentence reorder) | Game of Life | Algorithmic (matrix manipulation) |
> | :--- | :---: | :---: | :---: | :---: | :---: |
> | **Instruct** | 22.90 | 15.00 | 9.60 | 76.20 | 2.70 |
> | **Standard RLVR** | 12.40 | 17.00 | 9.80 | 75.10 | 3.90 |
> | **Long-horizon RL** | **47.10** | **22.50** | **18.80** | 74.90 | **4.20** |
>
> 3. We have now updated Table 2 with generalization results for our equal-compute baselines (Only-L1, Uniform-Mix, and Only-Long). Overall, these perform lower on harder math benchmarks compared to our method. We use Dr. GRPO [2] for all our baselines, including length 1 equal compute and length 1 during our curriculum. These can therefore be seen as independent baselines and perform lower than our method. As for improvements on harder benchmarks, we note that these happen without ever training on problems of the same difficulty level. We only ever train on long-horizon GSM8K problems generated from the GSM8K training set, and all our results are from the generalization that our training and data synthesis method offers. As such, gains on all these benchmarks as a whole do indicate substantial overall improvements.
>
>     &nbsp;
>
>     We would like to clarify that the claim, “while improvements obtained from RLVR on standard data is bounded by the base model’s capabilities, our method performs significantly better” is made in the context of Figure 3. We provide a clear justification of this in the paper (Section 4) and summarize it as follows: Yue et al., 2025 [2] show that RLVR on LLMs only improves the sample efficiency of reasoning capabilities already present in the base model, and no new capabilities are learnt. They observe that at a high pass@k (such as 128), capabilities of these RL trained models originate from and are bounded by the base model (with the pass@k performance quickly converging). Therefore, when an RL trained model is not bounded by the base/instruct model at high pass@k, one can empirically demonstrate that new skills are learnt. Our results on length 6, 7, and 8 GSM8K problems show that while standard length 1 RL training is bounded by the base model, our method exceeds both even at high pass@k - which is why we make that claim. As for compute matching, we ensure this by training our baselines up to the same number of training tokens seen under our h1 method. We checkpoint our baselines every 50 steps and report the best performing long-horizon GSM8K val-set checkpoint for all baselines.

---

> > ### Author Response · Authors · 2025-11-25
> > **Continuation of our response**
> >
> > Continuation of point 3:
> >
> > | Model/setting | MATH-500 | Symbolic P1 | Symbolic P2 | MMLU-Pro | AIME 2025 | AIME 2024 |
> > | :--- | :---: | :---: | :---: | :---: | :---: | :---: |
> > | **Instruct model** | 64.20 | 67.06 | 43.08 | 58.47 | 1.77 | 3.96 |
> > | *Standard RLVR on GSM8K* | | | | | | |
> > | **GSM8K RLVR** | 66.20 | 71.40 | 47.60 | 60.62 | 2.71 | 6.88 |
> > | *Equal compute training baselines* | | | | | | |
> > | **Only-L1** | 48.40 | 71.32 | 42.24 | 59.84 | 3.12 | 5.31 |
> > | **Uniform-Mix** | 64.40 | 64.48 | 39.16 | 60.22 | 2.50 | 5.28 |
> > | **Only-Long** | 65.60 | 72.18 | 47.52 | 60.71 | 1.72 | 6.46 |
> > | *Curriculum RL on Composed GSM8K Problems* | | | | | | |
> > | **Len-5 GSM8K** | **69.20** | **73.28** | **52.00** | **61.21** | **3.02** | **10.52** |
> >
> >
> > *Other concerns*
> > 1. Our compute matched baselines are trained using up to the same number of training tokens seen under our method, and the best checkpoints are chosen based on val-set accuracy. The data comes from the same training distribution (where Only-L1 refers to simply training on horizon $1$ problems, Only-Long refers to training only on composed horizon $5$ problems, and Uniform-Mix refers to training on the same problems as our method but without a curriculum i.e. shuffled). Therefore, the baselines are also trained up to an approximately similar number of training steps as our curriculum method. We have updated our paper to clarify this.
> >
> > 2. Shao et al. (2025) [3] show that spurious-reward gains are observed mainly on base/math specialized Qwen-2.5 models with strong pre-training priors, but not on Instruct models. In fact, when they apply RLVR with or without spurious rewards to the instruction-tuned variants of these Qwen Instruct models, they report minimal gains and suggest that these are harder to improve. This is consistent with our statement that improving Instruct models with RL is generally more difficult and that gains on Instruct checkpoints are more indicative of changes beyond standard instruction tuning, whereas gains on base models can often be attributed to eliciting capabilities already present from pretraining. Thank you for the opportunity to clarify this further.
> >
> > 3. We agree with the reviewer and have now updated the abstract when discussing our theory (from ‘achieves’ to ‘could achieve’). The theory in Appendix B is intended to provide an intuition as to why our method provides significantly better than single-step or direct long-horizon training, and to illuminate a connection between our method and dense rewards. We agree that it is not intended to completely characterize multi-step transformer reasoning, which would require making further assumptions about reasoning behaviors. We start with a simplified set up, as you note, and have now updated the paper to more clearly state our set up and assumptions. It should also be noted that self-correction and error cancellation would both raise the success rate, and as such our model can be viewed as analysing the worst case scenario, where error recovery is impossible. Incorporating these factors into our model would reduce sample complexity in all scenarios, but is likely to leave the exponential separation intact.
> >
> > 4. Our learning rate (between 2e-6 and 5e-6) is fairly consistent with prior RLVR/GRPO works such as DeepSeek R1 [4] and other important papers [5][6]. Overall, for our learning rates we did not run a full hyperparameter sweep and in practice observed ranges from 1e-6 to 5e-6 work well with stable improvements over several training steps and good generalization to in and out of domain benchmarks.
> >
> > 5. Our validation data is a composed set of separated out questions that are different from our training and test set. It comes from the original training problems (and the problems used in our val set are not in our train or test set). We see that validation and training performance broadly match and our results are indeed reported with the final checkpoints. It is still standard practice to check this against the validation set, but this confirms that the final checkpoint at each curriculum stage is useful.
> >
> > We sincerely appreciate the reviewer’s feedback and have aimed to address the comments above. We would be very grateful if the reviewer would consider raising their score.
> >
> > [1] REASONING GYM: Reasoning Environments for Reinforcement Learning with Verifiable Rewards, Stojanovski et al.
> >
> > [2] Understanding R1-Zero-Like Training: A Critical Perspective, Liu et al.
> >
> > [3] Spurious Rewards: Rethinking Training Signals in RLVR, Shao et al.
> >
> > [4] DeepSeek-R1: Incentivizing Reasoning Capability in LLMs via Reinforcement Learning, DeepSeek-AI et al.
> >
> > [5] Does Reinforcement Learning Really Incentivize Reasoning Capacity in LLMs Beyond the Base Model?, Yue et al.
> >
> > [6] Reinforcement Learning with Verifiable Rewards: GRPO's Effective Loss, Dynamics, and Success Amplification, Mroueh et al.

---

### Official Review · Reviewer_W3rn · 2025-10-31

**Soundness:** 3
**Presentation:** 3
**Contribution:** 3
**Rating:** 4
**Confidence:** 5

**Summary:**

The proposed curriculum-based RL method, trained on simple 6th-grade math problems (GSM8K), dramatically improves performance on challenging, unseen Olympiad-level benchmarks (GSM-Symbolic, MATH-500, AIME), achieving up to 2.65× accuracy gains. These long-horizon reasoning improvements persist and even exceed baselines at high pass@k settings, demonstrating that RL enables models to discover entirely new reasoning pathways.
Theoretical Insight:
The authors prove that curriculum-based reinforcement learning with outcome rewards provides formal guarantees for effective generalization in reasoning tasks.

**Strengths:**

The method yields up to 2.65× accuracy gains on Olympiad-level benchmarks (GSM-Symbolic, MATH-500, AIME). Long-horizon improvements outperform baselines even at high pass@k, proving that RL enables models to learn entirely new reasoning paths.

**Weaknesses:**

1.Long-context is common to improve the performance of LLMs, the propsoed noviely is not obvious in the direction of long-context.
2.The improvement is not obvious for AIME24, AIME25.
3.The experiments on one-single LLM, there is not strong proof if there is not improvements on the other LLMs such as r1-distill, and other reasoning LLMs.

**Questions:**

1.Given that long-context training is a common technique for improving LLM performance, what specific novel contributions does the proposed method introduce in the long-context direction?
2.Why are the performance improvements not substantial on challenging benchmarks such as AIME24 and AIME25?
3.Why are experiments conducted on only a single LLM, and why is there no evaluation on other reasoning-focused models such as r1-distill to demonstrate generalizability?

---

> ### Author Response · Authors · 2025-11-25
>
> Dear Reviewer,
>
> Thank you for your time and valuable feedback! We appreciate the opportunity to address the points raised and further improve our work.
>
> 1. We would like to clarify that long-context training involves providing LLMs with a large input context and testing the ability to recall or extract information. Our work (long-horizon reasoning) is focused on long outputs, which are required to solve challenging multi-step CoT reasoning problems. For this direction, we provide a clear definition and a method that involves constructing a synthetic RL curriculum that progressively increases in complexity based on compositions and transfers very well to several harder in-domain math benchmarks along with long-context benchmarks. We compare this against other work in long-context and length generalization in our related work section and Appendix A (with an entire subsection devoted to long-context) and show how our work is novel and contributes towards a currently understudied research direction. Overall, we show that long-horizon performance can be improved without needing new data and therefore aim to drive progress towards this newer direction.
>
> 2. Our models go from 3.96% to 10.52% on AIME 2024 (avg@32 for reduced variance), which we believe is substantial. We never train on AIME-level problems and are still able to transfer with just training on 6th grade mathematics data under a composed synthetic curriculum. Moreover, we also test on **MATH, MMLU Pro, Symbolic P1, Symbolic P2, LongBench-v2, and Hash-hop.** These are used as benchmarks for frontier models and are considered very challenging, and we are able to show generalized improvements to each and every benchmark on this list. Furthermore, in order to show even broader generalization, we study the empirical performance of our long-horizon trained models on several **ReasoningGym** [1] domains. ReasoningGym consists of a diverse set of (new) reasoning environments that allow us to evaluate cross-domain transfer and skill generalization. We test performance on propositional logic, graphs, algorithmic (sentence reordering), and algorithmic (matrix operations) and see substantial transfer beyond math and long-context domains. We provide our results below and have also updated Section 5 of our paper to discuss cross-domain generalization in more detail.
> | Model / setting | Propositional logic | Graphs (largest island) | Algorithmic (sentence reorder) | Game of Life | Algorithmic (matrix manipulation) |
> | :--- | :---: | :---: | :---: | :---: | :---: |
> | **Instruct** | 22.90 | 15.00 | 9.60 | 76.20 | 2.70 |
> | **Standard RLVR** | 12.40 | 17.00 | 9.80 | 75.10 | 3.90 |
> | **Long-horizon RL** | **47.10** | **22.50** | **18.80** | 74.90 | **4.20** |
>
> 3. In our work, we test our method on two model families (Qwen and Llama - see Appendix E). We see consistent improvements across the benchmarks tested for both model families. We fully agree about showing generalized reasoning performance on other domains and have now studied the performance of our models on ReasoningGym domains. We have provided this above, and overall see substantial transfer to harder math benchmarks, long-context domains, and general multi-step reasoning problems from only training on composed long-horizon 6th grade math problems.
>
> We sincerely appreciate the reviewer’s feedback and have aimed to address all the comments above. We would be very grateful if the reviewer would consider raising their score further.
>
> [1] REASONING GYM: Reasoning Environments for Reinforcement Learning with Verifiable Rewards, Stojanovski et al.

---

### Official Review · Reviewer_TFsz · 2025-11-02

**Soundness:** 3
**Presentation:** 3
**Contribution:** 3
**Rating:** 4
**Confidence:** 4

**Summary:**

This paper introduces a novel curriculum-based reinforcement learning framework for improving long-horizon reasoning capabilities in large language models. The key innovation involves synthetically composing short-horizon problems into complex dependency chains of arbitrary length, then training models using outcome-only rewards under a progressive curriculum that automatically increases complexity. The method demonstrates strong empirical results, achieving improvement on challenging benchmarks like AIME while training only on composed GSM8K problems. Theoretical analysis shows curriculum-based RL achieves exponential improvement in sample complexity over full-horizon training.

**Strengths:**

The method provides a scalable path for improving long-horizon reasoning without requiring new annotations.
Demonstrates significant improvements on challenging benchmarks.

**Weaknesses:**

The core methodology bears significant resemblance to "Training Large Language Models for Reasoning through Reverse Curriculum Reinforcement Learning" (R³). Both approaches:
Use curriculum learning over reasoning steps
Employ outcome-only rewards
Gradually increase difficulty from simple to complex problems
Focus on improving multi-step reasoning capabilities
While the current work uses problem composition rather than demonstration sliding, the fundamental curriculum RL framework appears quite similar. The paper would benefit from a more explicit comparison and differentiation.

**Questions:**

Could you elaborate on how your approach fundamentally differs from R³'s reverse curriculum RL? What specific advantages does problem composition offer over demonstration-based curriculum learning?
The method shows strong transfer from GSM8K to math benchmarks, but how would it perform on non-mathematical reasoning tasks (e.g., logical deduction, commonsense reasoning)?

---

> ### Author Response · Authors · 2025-11-25
>
> Dear Reviewer,
>
> Thank you for your time and valuable feedback! We appreciate the opportunity to address the points raised and further improve our work.
>
> 1. Thank you for providing a reference to R^3 [1], we have now included a discussion in our updated Related Work Section. While there are some similarities around using a curriculum, our work can be seen as specifically solving two major limitations of R^3: the requirement for step-level demonstrations; and the inability to extend the horizon length beyond what is included in the dataset.
>
>     &nbsp;
>
>     The method employed in [1] works by collecting a dataset with step-level demonstrations, and then training via a curriculum that progressively *removes* later steps, forcing the model to solve more steps. However, collecting step-level demonstrations can be costly and time-consuming, and the maximum (step) horizon length trained to via this method is limited by the maximum number of demonstration steps in any sample in the dataset.
>
>     &nbsp;
>
>     By contrast, h1 works by combining short-horizon data with only final-answer labels. This allows it to be used with far more datasets, as there is no requirement to have/collect demonstration data. It also allows us to construct data with potentially unbounded horizon length: given a chain of n steps, we can create training data with n+1 steps by simply appending an entirely new problem (that depends on the previous solutions). Therefore, existing problems can be combined by composing them to create long horizon problems of any length. This allows for entirely programmatic generation of synthetic data of unbounded length, where the problems themselves become one step of the chain. We have now endeavored to make this distinction clear between our work and [1] in an updated version of our paper.
>
>
> 2. We have updated our paper with new results, testing how well our models trained on 6th grade long-horizon problems using our method generalize to broader non-mathematical reasoning tasks. We test our models on several ReasoningGym [2] domains, which evaluate skills such as maintaining a working memory, graph traversal, rule following, and general multi-step reasoning. ReasoningGym consists of a diverse set of (new) reasoning environments that allow us to evaluate cross-domain transfer and skill generalization. We test performance on propositional logic, graphs, algorithmic (sentence reordering), and algorithmic (matrix operations) and see substantial transfer beyond math and long-context domains. We provide our ReasoningGym results below and have updated Section 5 in our paper.
> | Model / setting | Propositional logic | Graphs (largest island) | Algorithmic (sentence reorder) | Game of Life | Algorithmic (matrix manipulation) |
> | :--- | :---: | :---: | :---: | :---: | :---: |
> | **Instruct** | 22.90 | 15.00 | 9.60 | 76.20 | 2.70 |
> | **Standard RLVR** | 12.40 | 17.00 | 9.80 | 75.10 | 3.90 |
> | **Long-horizon RL** | **47.10** | **22.50** | **18.80** | 74.90 | **4.20** |
>
> We sincerely appreciate the reviewer’s feedback and have aimed to address all the comments above, including testing transfer to non-mathematical domains with significant improvements. We would be very grateful if the reviewer would consider raising their score further.
>
> [1] Training Large Language Models for Reasoning through Reverse Curriculum Reinforcement Learning, Xi et al.
>
> [2] REASONING GYM: Reasoning Environments for Reinforcement Learning with Verifiable Rewards, Stojanovski et al.

---

> > ### Comment · Reviewer_TFsz · 2025-11-27
> >
> > The authors have resolved my concerns, and I have raise the score.

---

### Official Review · Reviewer_t28Z · 2025-11-04

**Soundness:** 2
**Presentation:** 3
**Contribution:** 2
**Rating:** 4
**Confidence:** 3

**Summary:**

The paper proposes a way to teach language models long-horizon reasoning using chained problems from datasets like GSM8k. The authors synthetically chain simple math problems into multi-step dependency chains, then train an LLM with RLVR under a curriculum, which gradually increase the chain length. This curriculum helps the model learn not just to solve individual steps but to reliably carry intermediate results forward through long chains of thought. The experiments show that model trained this way on composed GSM8k questions can also achieve gains on harder benchmarks like MATH-500 and AIME. The paper also claims that curriculum-based RL with outcome reward can yield exponential sample complexity improvement over full-horizon training.

**Strengths:**

* The paper's main idea, which systematically compose existing short-horizon problems into explicit long-horizon chains and then doing RL with it seems fresh.They way they formalize long-horizon difficulty via both per-step reliability and "context management" seems nontrivial.
* The experiment shows a set of ablations comparing horizon curricula (Len-2 --> Len-5) against a few baselines like only-l1, only-long, showing that their proposed curriculum based training is effective. They also show results on harder benchmarks like AIME to show the generalization of their proposed method. The authors also provide a sample-complexity analysis of curriculum vs full-horizon RL, which could potentially helps ground the empirical design.
* The paper is generally well written and easy to follow. The method section walks through atomic tasks, adapters and composition with a human-readable example.
* The work addresses a practical bottleneck: there is lots of verifiable short-horizon data but very little high-quality long-horizon data, especially for RL on LLMs. The paper offers a concrete recipe for turning that short-horizon data into a scalable RL curriculum and backs it with evidence that this actually produces meaningful new capabilities

**Weaknesses:**

* The models and datasets used in the experiment seems toyish. A Qwen2.5-3B model is small compared with other mainstream models. The experiment only includes such a small model, which may weaken the contribution of the proposed method. The dataset's scale is also small. It is unclear if the proposed method can easily scale up to and works on larger datasets (e.g., NuminaMath).
* The only non-math results are on long-context reading benchmarks (LongBench-v2, Hash-hop), which are used purely for evaluation. That makes it hard to know whether the method really teaches a general long-horizon skill, or just exploits very specific algebraic structure plus better “carry the number” habits.
* The paper’s related work mostly focuses on RLVR, length generalization, and generic curriculum RL, but doesn’t clearly separate what’s really new in this composition + horizon-curriculum recipe vs these.
* Since the paper claims to avoid inference-time search/PRMs and “train the model to internalize long-horizon reasoning structures,” it would help to compare against a strong search-based method in the experiment.
* The paper leans heavily on strong numbers like “2.65x” improvement on AIME 2024 and good gains on MATH-500, GSM-Symbolic, LongBench-v2, and Hash-hop. However, AIME benchmarks are tiny (dozens of problems), so going from 3.96% --> 10.52% may correspond to only a small number of additional questions solved. For long-context tasks, improvements are modest (e.g., 35.3 --> 37.9 on LongBench-v2) and there isn’t much qualitative analysis of what changed.
* The proposed theory assumes relatively clean factorization of errors and horizon structure that may not hold in messy LLM trajectories. The paper assumes that correctness at each step can be summarized by one Bernoulli variable $q_j$, and long-horizon correctness is exactly the product of those. In real LLM trajectories, there are things like error cancellation, redundant reasoning, or later self-correction. Those cases could break the assumptions made by this paper.

**Questions:**

See above.

---

> ### Author Response · Authors · 2025-11-25
>
> Dear Reviewer,
>
> Thank you for your time and valuable feedback! We appreciate the opportunity to address the points raised and further improve our work.
>
> 1. Both Qwen-2.5-3B-Instruct and Llama-3.2-3B are standardized and widely used models (more than 8 million HF downloads in the past month). Our training data is based on long-horizon GSM8K problems but it transfers significantly not only in-domain, but also to AIME, GSM-Symbolic, MATH, MMLU-Pro, LongBench-v2, and Hash-hop. These are used as benchmarks for much larger (frontier) models too, and our method translates to gains on all of them. Furthermore, in Appendix D we present early results with Qwen-2.5-7B-Instruct trained on chained MATH problems rather than GSM8K.
>
> 2. We completely agree that it is important to study other non-math benchmarks. We now study empirical performance of our long-horizon trained model on several ReasoningGym [1] domains. ReasoningGym consists of a diverse set of (new) reasoning environments that allow us to evaluate cross-domain transfer and skill generalization. We test performance on propositional logic, graphs, algorithmic (sentence reordering), and algorithmic (matrix operations) and see substantial transfer beyond math and long-context domains. We provide our ReasoningGym results below and have updated Section 5 in our paper.
> | Model / setting | Propositional logic | Graphs (largest island) | Algorithmic (sentence reorder) | Game of Life | Algorithmic (matrix manipulation) |
> | :--- | :---: | :---: | :---: | :---: | :---: |
> | **Instruct** | 22.90 | 15.00 | 9.60 | 76.20 | 2.70 |
> | **Standard RLVR** | 12.40 | 17.00 | 9.80 | 75.10 | 3.90 |
> | **Long-horizon RL** | **47.10** | **22.50** | **18.80** | 74.90 | **4.20** |
>
> 3. We are among the first works to build a synthetic RL curriculum that increases in complexity, uses compositions to do so, and obtains generalization across domains. Independently, curriculum learning has been studied, and RLVR has also received significant attention. We have listed prior work (across length generalization and RLVR) in our Related Work section and Appendix A; however, there has been limited focus on building synthetic curricula using existing data for reinforcement learning transfer across long-horizon problems. Length generalization mostly focuses on small-scale tasks, such as addition and multiplication, but does not focus on out-of-domain transfer, reinforcement learning-based post-training, or tasks that are not purely algorithmic.
>
> 4. In Figure 3, we provide a comparison between our models and standard RLVR/Instruct models even at high pass@k. Pass@k can be seen as the strongest possible inference-spend baseline, against which we still show substantial improvements. Additionally, we’re now providing another baseline that uses tree-of-thought [2] to spend more inference-compute and compare against our training method below. We have also updated Table 1 with this baseline.
> | Model / setting | L-1 | L-2 | L-3 | L-4 | L-5 | L-6 | L-7 | L-8 |
> | :--- | :---: | :---: | :---: | :---: | :---: | :---: | :---: | :---: |
> | **Tree of Thought** | 83.30 | 39.40 | 13.30 | 2.00 | 0.00 | 0.00 | 0.00 | 0.00 |
> | **Our Method (Len-5 H1)** | 85.97 | 58.51 | 36.39 | 18.77 | 9.82 | 3.53 | 3.17 | 2.22 |
>
> 5. Our method shows non-trivial improvements on MATH-500, GSM Symbolic P1, GSM Symbolic P2, MMLU Pro Math, AIME 2024, AIME 2025, LongBench-v2, Hash-hop, and now several ReasoningGym domains without ever training for these tasks. While we agree that AIME is a smaller benchmark, we test at an average of 32 samples and the results are supplemented by other benchmarks. We have also updated our paper to provide a clear qualitative example and analysis in Appendix E.
>
> 6. The theory in Appendix B is intended to provide an intuition as to why our method works significantly better than single-step or direct long-horizon training, and to show a connection between our method and dense rewards. We agree that it is not intended to completely characterize multi-step transformer reasoning, which would require making further assumptions about reasoning behaviors. We start with a simplified set up, as you note, and have now updated the paper to more clearly state our set up and assumptions. It should also be noted that self-correction and error cancellation would both raise the success rate, and as such our model can be viewed as analyzing the worst case scenario, where error recovery is impossible. Incorporating these factors into our model would reduce sample complexity in all scenarios, but is likely to leave the exponential separation intact.
>
> We sincerely appreciate the reviewer’s feedback and have aimed to address all the comments above. We would be very grateful if the reviewer would consider raising their score further.
>
> [1] REASONING GYM: Reasoning Environments for Reinforcement Learning with Verifiable Rewards, Stojanovski et. al.
> [2] Tree of Thoughts: Deliberate Problem Solving with Large Language Models, Yao et. al.

---

### Comment · Area_Chair_G7w6 · 2025-11-26

Dear Reviewers,

Would you please check authors' rebuttal and see if they have addressed your comments?

Best

AC

---

### Meta-Review · Area_Chair_zYui · 2026-01-06

**Summary:**

This submission receives ratings of 4, 4, 4, 4. The major contribution of this work is the idea of transforming short problems into long-horizon curriculum for RL training. The reviewers raises questions of whether the presented work demonstrates sufficiently general and clear contributions to this topic. The consensus concern among reviewers is the mismatch between the claims and empirical demonstration. First, the absolute performance of the models on relatively simple (not quite long-horizon tasks) benches AIME and GSM are not very promising, and the delta that the proposed technical brings is also relatively minor --- making it difficult to support the claims on "Reason over Longer Horizons". Second, the reviewers raise questions on its conceptual advancements of its curriculum RL strategy. Finally, the evaluation can be improved in terms of model diversity, baselines.

**Reviewer Concerns:**

Concerns that are mostly addressed during rebuttal:
1) Difference between the proposed curriculum RL and R^3.
2) Add non-math Reasoning Gym results, also for the generalization issue.
3) Clarification between long context and long horizon tasks.
4) soft claims in theoretical analysis.

Those that are not fully addressed:
1) Absolute gains on AIME and LongBench are limited;
2) unconvinced evidence to support claims of new reasoning capabilities.
3) the conceptual advancement of the presented RL technique.

**Reviewer Scores:**

All four reviewers gave initial ratings of 4, 4, 4, 4. One reviewer (tfsz) will raise the rating to a positive one (e.g., 6). That is a chance that reviewer t28z would raise to 6, and it is likely that the other two reviewers would not change their ratings given the responses.

---

### Decision · Program_Chairs · 2026-01-26

Reject